# Perception of social interaction compresses subjective duration in an oxytocin-dependent manner

Rui Liu[1,2,3,4†], Xiangyong Yuan[2,3,5†], Kepu Chen[1,2,3†], Yi Jiang[2,3,5*], Wen Zhou[1,2,3*]

[1]CAS Key Laboratory of Behavioral Science, Institute of Psychology, Chinese Academy of Sciences, Beijing, China; [2]Department of Psychology, University of Chinese Academy of Sciences, Beijing, China; [3]CAS Center for Excellence in Brain Science and Intelligence Technology, Shanghai, China; [4]Donders Institute for Brain, Cognition and Behaviour, Radboud University, Nijmegen, Netherlands; [5]State Key Laboratory of Brain and Cognitive Science, Institute of Psychology, Chinese Academy of Sciences, Beijing, China

**Abstract** Communication through body gestures permeates our daily life. Efficient perception of the message therein reflects one's social cognitive competency. Here we report that such competency is manifested temporally as shortened subjective duration of social interactions: motion sequences showing agents acting communicatively are perceived to be significantly shorter in duration as compared with those acting noncommunicatively. The strength of this effect is negatively correlated with one's autistic-like tendency. Critically, intranasal oxytocin administration restores the temporal compression effect in socially less proficient individuals, whereas the administration of atosiban, a competitive antagonist of oxytocin, diminishes the effect in socially proficient individuals. These findings indicate that perceived time, rather than being a faithful representation of physical time, is highly idiosyncratic and ingrained with one's personality trait. Moreover, they suggest that oxytocin is involved in mediating time perception of social interaction, further supporting the role of oxytocin in human social cognition.
DOI: https://doi.org/10.7554/eLife.32100.001

*For correspondence:
yijiang@psych.ac.cn (YJ);
zhouw@psych.ac.cn (WZ)

†These authors contributed equally to this work

**Competing interests:** The authors declare that no competing interests exist.

## Introduction

As a joke, Albert Einstein once gave this picture to explain his theory of relativity: "Put your hand on a hot stove for a minute, and it seems like an hour. Sit with a pretty girl for an hour, and it seems like a minute. That's relativity." This seemingly intuitive picture has no bearing on the structure of space-time, yet nicely illustrates the now established finding that mental time deviates, sometimes significantly, from physical time (*Eagleman, 2008*).

To date, the deviation between our experienced time and the physical time has mostly been attributed to sensory properties of the external stimuli (*Eagleman, 2008*) and their context (*Shi et al., 2013*). It has been proposed that subjective time is 'warped' by the neural energy involved in representing sensory inputs (*Eagleman and Pariyadath, 2009*; *Zhou et al., 2018*). For instance, intense and/or moving stimuli are generally experienced as longer in duration (*Fraisse, 1984*) as they evoke stronger perceptual responses in cortical neurons (*Mayo and Sommer, 2013*). Little is known as to what role, if any, we the experiencers play in the time we experience. Considering our gregarious nature and the ubiquity of social interactions in daily life, we set out to probe this issue by examining time perception of social interactions and the inter-individual differences therein.

**eLife digest** Einstein once joked: "Put your hand on a hot stove for a minute, and it seems like an hour. Sit with a pretty girl for an hour, and it seems like a minute. That's relativity." While it may not have helped explain the space-time continuum, his joke neatly captures how time can appear to pass at different rates. This perception depends in part on the sensory properties of the stimuli we are experiencing. Intense stimuli, such as bright and fast-moving objects, trigger stronger responses in the brain than less intense stimuli, and so we perceive them as longer lasting.

But what role do we, as the experiencers, play in how we perceive time? To find out, Liu, Yuan, Chen et al. showed volunteers pairs of movie clips, each featuring two human figures outlined by dots. In one clip, the two figures interacted socially, for example by passing an object between them. In the other, the two figures moved independently of each other. The volunteers had to decide which clip lasted longer.

The volunteers generally judged clips containing social interactions to be shorter than those without such interactions, even when this was not the case. Moreover, volunteers with better social skills tended to underestimate the length of the social interaction clips to a greater extent.

Previous studies have shown that people who are more social tend to have higher levels of a hormone called oxytocin in their blood. Oxytocin is sometimes referred to as the 'love hormone' because it promotes social behavior and bonding. Applying an oxytocin nasal spray to the volunteers who were less socially proficient caused them to perceive the social interaction clips as shorter than before. By contrast, socially proficient volunteers who used a nasal spray that blocks the effects of oxytocin perceived these clips as longer than they had done previously (although they still judged the clips to be shorter than videos that did not show people interacting).

The perception of time thus varies between people and may depend at least in part on personality. These results open up a new avenue for studying and manipulating how we process social situations. This could eventually benefit people who struggle with social interactions, such as those with autism spectrum disorders.

DOI: https://doi.org/10.7554/eLife.32100.002

An important medium of social interaction is body gestures, from which most humans efficiently extract others' attitudes and intentions even when the gestures are portrayed by only a handful of point lights attached to the head and major joints (*Johansson, 1973*). Such efficacy is considered evolutionarily endowed — Human newborns and infants exhibit a predisposition to attend to the motions of biological entities (i.e. biological motion)(*Fox and McDaniel, 1982*; *Simion et al., 2008*); and the perception of biological motion engages a specific network of distributed neural areas, particularly the superior temporal sulcus (STS) (*Grossman et al., 2005*; *Grossman and Blake, 2002*; *Vaina et al., 2001*) that plays an important role in social perception in both monkeys and humans (*Allison et al., 2000*). Meanwhile, inter-individual variation is noteworthy. People with autism characterized by impaired social interaction and communication show both a deficit of biological motion perception (*Blake et al., 2003*; *Klin et al., 2009*) and abnormalities in the STS (*Zilbovicius et al., 2006*). Social proficiency varies widely even among neurotypical individuals, and is manifested behaviorally as a stable personality trait (*Digman, 1990*). This inter-individual variance has been associated with plasma concentrations of oxytocin, a well-documented prosocial neuropeptide, as well as polymorphisms of its receptor gene OXTR (*Andari et al., 2014*; *Donaldson and Young, 2008*; *Modahl et al., 1998*; *Parker et al., 2014*; *Skuse et al., 2014*; *Tost et al., 2010*). Likely through interactions with endogenous oxytocin, intranasally administered oxytocin (*Dal Monte et al., 2014*; *Freeman et al., 2016*; *Lee and Weerts, 2016*; *Striepens et al., 2013*) is found to alter the processing of social stimuli including biological motion in manners that depend on one's social proficiency (*Bartz et al., 2011*; *Kéri and Benedek, 2009*) as well as blood oxytocin concentration (*Parker et al., 2017*).

There has been limited research on the perceptual processing of social interaction between biological entities, particularly as depicted in biological motion displays. Nonetheless, social interaction is far beyond the movements of individuals. In its simplest form, it involves two agents acting in a meaningful manner: one agent executes a gesture, the other recognizes it and acts accordingly.

Temporally adjacent actions generally tend to be inferred as a causal sequence and hence communicative (*Lagnado and Sloman, 2006*). But ultimately that social 'meaning' is derived from the observer's interpretations of the agents' actions and the relationship in between, and is, to a certain degree, subjective in nature.

The present study aimed to address the effect of perceived social interaction on subjective time and its relationship with one's intrinsic social proficiency. In a series of experiments, we carefully manipulated point-light displays of acting agents to dissociate the perception of social interaction and that of biological motion. We assessed temporal perception of such displays in individuals varying in social proficiency, and critically examined the role of oxytocin in this process. Given the aforementioned relationships among social proficiency, oxytocin and neural processing of social stimuli, our hypothesis was that both social proficiency, which reflects endogenous oxytocin level, and exogenous manipulations of oxytocin level would influence the neural efficacy in processing social interactions, and thereby modulate the subjective time of perceived social interactions.

## Results

### Perception of social interactions portrayed by point-light displays shortens subjective duration in a manner dependent on the observer's social proficiency

As an initial step to qualify the influence of perceived social interaction on subjective duration, we selected 10 point-light displays of motion sequences from the Communicative Interaction Database (*Manera et al., 2010*), each portraying two agents engaging in a communicative interaction that usually involved an object (triadic interaction, see Materials and methods), and made from them an essentially physically matched set of 10 noncommunicative motion sequences by cross-pairing the agents from different interactions (see *Supplementary file 1*). Observers were shown two motion sequences in each trial — one communicative, the other noncommunicative, one after the other in random order — and were asked to report via button press which (the first or the second) appeared longer in duration (*Figure 1*). We kept the duration of one motion sequence fixed at 1000 ms (communicative or noncommunicative, each in 50% of trials in random order), and varied the duration of the other one from trial to trial (from 400 to 1600 ms). In different blocks, the two motion sequences were either both shown upright, or both inverted. By assessing which motion sequence observers perceived as being longer in duration, we obtained psychometric curves that depicted the probability of choosing the communicative over the noncommunicative as a function of their physical duration difference (communicative – noncommunicative). The duration difference corresponding to a probability of 50% marks the point of subjective equality (PSE), which would be around 0 if there is no influence of social interaction on time perception. Half the interquartile range of the fitted psychometric function marks difference limen, an index of one's sensitivity in temporal perception.

Twenty-four observers (12 females) performed the duration judgment task in Experiment 1. In the upright condition, the mean PSE was 69.3 ms, significantly above 0 ($t_{23}$ = 5.60, p<0.001, Cohen's d = 1.14, *Figure 2A*). In other words, an upright communicative motion sequence compressed subjective duration such that it needed to be 69.3 ms longer to be perceived as equal in duration to an upright noncommunicative motion sequence. In the inverted condition, by contrast, the mean PSE was significantly smaller ($t_{23}$ = −5.82, p<0.001, Cohen's d = −1.19) and not different from 0 ($t_{23}$ = −1.51, p=0.15) (*Figure 2A*). Inversion is known to impair the perception of biological motion and thereby social interactions mediated by biological motion, yet does not affect low-level visual features (*Troje and Westhoff, 2006*). This result hence verified that the temporal compression effect of perceived social interactions could not be due to low-level non-biological visual features. The data from individual observers conformed with the averaged patterns. They are summarized in *Figure 2D*, in which each observer's PSEs from the upright (y-axis) and the inverted (x-axis) conditions are plotted against each other and shown as a red dot. Most red dots fell above the dashed line of slope 1, although several of them were close to the dashed line; that is, the observers were largely biased towards perceiving the communicative motion sequences as shorter in duration than the noncommunicative ones when the agents were shown upright as opposed to upside down, despite that the task did not require explicit social processing. Their temporal sensitivities remained

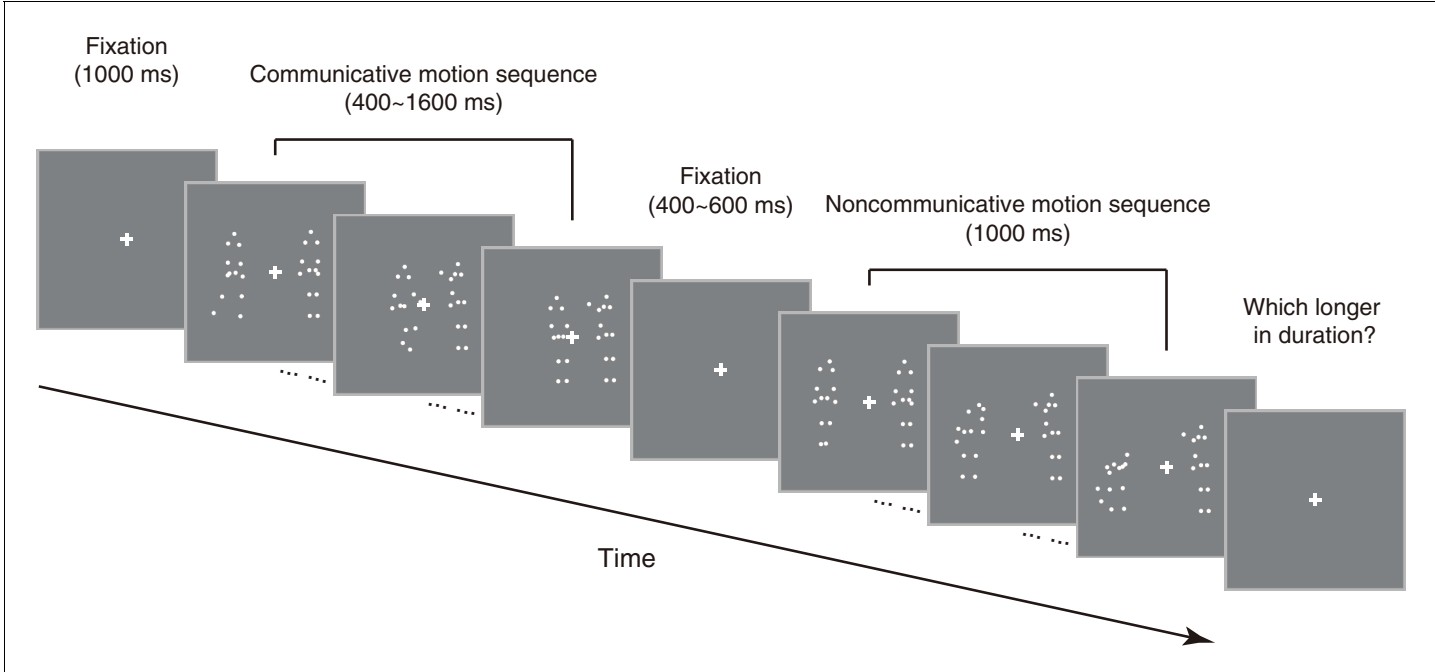

**Figure 1.** Schematic illustration of an exemplar trial in the duration judgment task.
DOI: https://doi.org/10.7554/eLife.32100.003

unchanged between the two conditions, as indicated by a comparison of the difference limens ($t_{23}$ = 0.70, p=0.49).

The point-light trajectories of two agents acting communicatively (coordinately) could be spatially and temporally more correlated than those of two agents acting noncommunicatively (independently) (*Bassili, 1976*; *Chartrand and Bargh, 1999*). Extraction of such a spatial-temporal relationship between two upright agents, rather than the social meaning per se, could have caused the observed temporal compression effect. To address this possibility, we conducted Experiments 2 and 3, where we eliminated the social aspect of the original communicative displays while keeping the spatial-temporal pattern differences between the communicative and the noncommunicative motion sequences unchanged. In Experiment 2, this was done by inserting a temporal lag of 700 ms in between every two acting agents (upright and inverted, communicative and noncommunicative) (*Manera et al., 2013*). In Experiment 3, we spatially swapped the two agents in each display such that they faced in opposite directions instead of facing each other. The two experiments were otherwise identical to Experiment 1. Analyses of the results from 48 observers (24 in each of Experiments 2 and 3; 27 females) indicated that neither the temporally delayed 'communicative' motion sequences nor the spatially swapped ones altered temporal perception relative to their 'noncommunicative' counterparts (*Figure 2B and C*). The PSEs were not significantly different from 0 regardless of whether the motion sequences were presented upright ($t_{23}$s = 1.39 and 0.34, ps = 0.18 and 0.74, for Experiments 2 and 3, respectively) or upside down ($t_{23}$s = 0.01 and −0.14, ps = 0.99 and 0.89, respectively). Between the upright and the inverted conditions, there was no significant difference in PSE ($t_{23}$s = −1.02 and −0.28, ps = 0.32 and 0.78, for Experiments 2 and 3, respectively) or difference limen ($t_{23}$s = −0.07 and 0.84, ps = 0.95 and 0.41, respectively). We also examined individual data and plotted each observer's PSEs from the upright and the inverted conditions against each other. As shown in *Figure 2D*, the values fell on both sides of the dashed line of slope 1 and were centered around the origin for both Experiments 2 (blue squares) and 3 (lime triangles). Moreover, an omnibus ANOVA of the pooled PSEs across Experiments 1 to 3 confirmed a significant interaction between the vertical orientation of the agents (upright or inverted) and experiment (Experiment 1, 2, or 3) ($F_{2, 69}$ = 9.98, p<0.001, Cohen's f = 0.54). No such interaction was found with the difference limens ($F_{2, 69}$ = 0.23, p=0.80). We hence concluded that the temporal compression effect observed in Experiment 1 was absent in Experiments 2 and 3. The mere spatial-temporal correlation between the

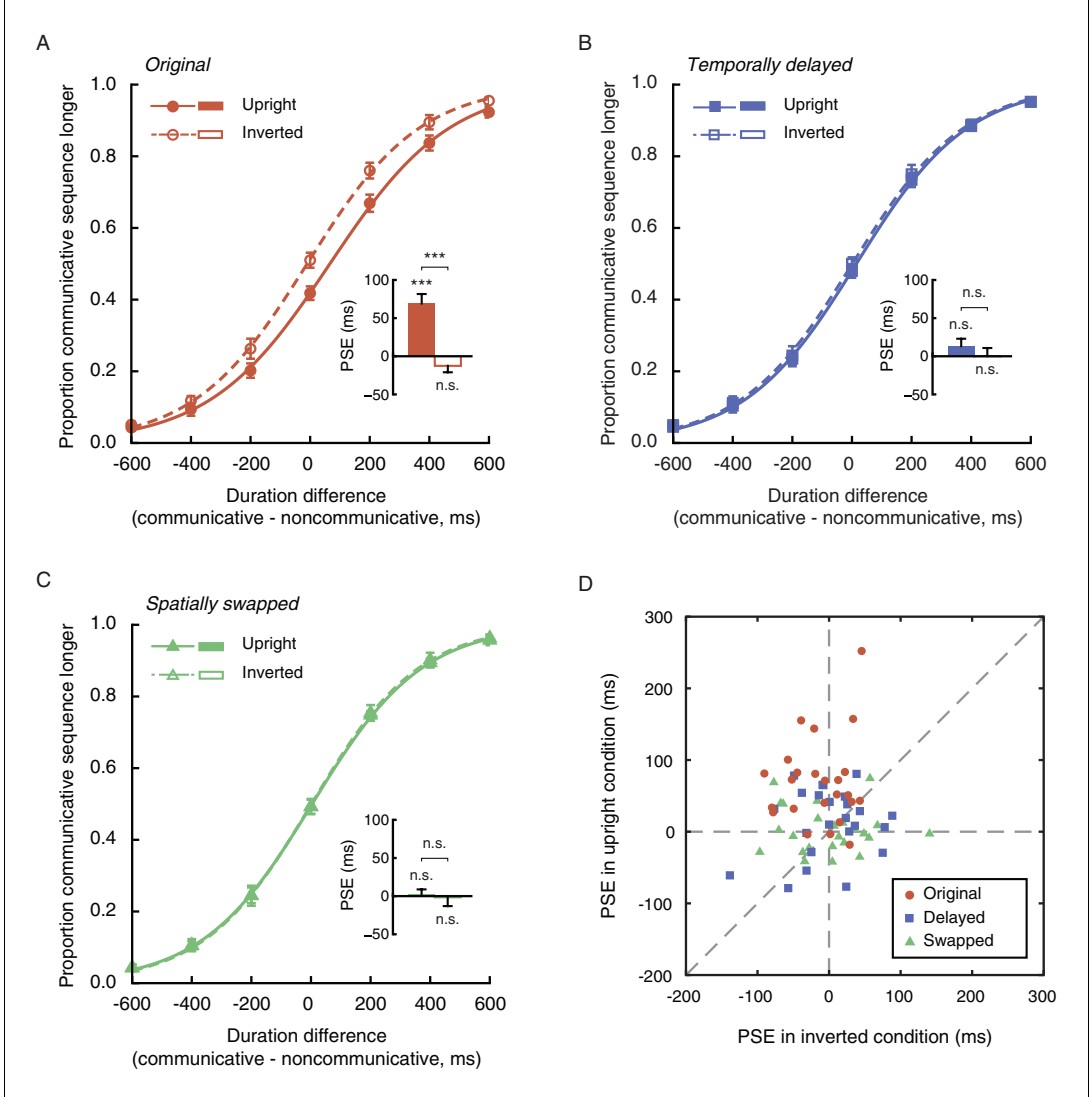

**Figure 2.** Perception of social interaction shortens subjective duration. (**A**) Proportion of responses in which observers reported a communicative motion sequence as longer in duration than a noncommunicative one, plotted as a function of the physical duration difference between the two. Data are shown for the upright (solid curve) and the inverted (dashed curve) conditions in Experiment 1. Inset shows the PSEs. A PSE of 0 indicates a consistency between subjective duration and physical duration. (**B-C**) Psychometric functions for Experiments 2 and 3 where a temporally delayed version (**B**) and a spatially swapped version (**C**) of the motion sequences in Experiment 1 were respectively used. In both cases, the strengths of the spatial-temporal correlations between acting agents were unaltered but the communicative intentions were disrupted. (**D**) The PSEs for the upright condition versus the PSEs for the inverted condition for individual observers in Experiments 1 (red dots), 2 (blue squares) and 3 (lime triangles). A slope of 1 (dashed diagonal line) represents comparable PSEs for the upright and the inverted conditions. Error bars: standard errors of the mean; ***p<0.001.

DOI: https://doi.org/10.7554/eLife.32100.004

The following figure supplement is available for figure 2:

**Figure supplement 1.** Supplementary Experiment.
DOI: https://doi.org/10.7554/eLife.32100.005

moves of two agents, without a recognizable communicative intention (disrupted by the temporal and spatial manipulations in Experiments 2 and 3), was insufficient to affect temporal perception. We further verified in a supplementary experiment (24 observers; 12 females) using Michotte-like launching and streaming events (between two objects) (*Michotte et al., 1963*) that the contingency between the movements of two entities, or the inference of causality, could not account for the

temporal compression effect associated with the perception of social interactions (*Figure 2—figure supplement 1*).

On the other hand, the recognition of communicative intention is unlikely to spontaneously occur in all observers. Social communicative ability has been shown to be a continuum that extends from patients with autism into the neurotypical population (*Baron-Cohen, 1995*; *Frith, 1991*; *Nummenmaa et al., 2012*; *von dem Hagen et al., 2011*). The degree of autistic traits (or lack of social proficiency), as measured by the Autism Spectrum Quotient (AQ), varies substantially even among healthy young adults. Such variance is particularly pronounced in males, who generally score higher than females on the AQ (*Baron-Cohen et al., 2001*). We wondered if the extent of temporal compression induced by the perception of social interactions (see *Figure 2D* for inter-individual differences) was a manifestation of one's social proficiency. To this end, we recruited a larger sample of 90 male observers and carried out Experiment 4, which employed the same task as Experiment 1 except that all motion sequences were presented upright (i.e., the inverted condition that served as a control in Experiment 1 was not included). Each observer's autistic-like tendency was also assessed with the AQ. Overall, Experiment 4 replicated the temporal compression effect observed in Experiment 1. The mean PSE was 38.5 ms, comparable to that of the male observers in Experiment 1 (45.8 ms) and significantly above 0 ($t_{89} = 6.07$, p<0.001, Cohen's d = 0.64, *Figure 3A*). Critically, inspection of the individual data revealed a significant negative correlation between PSE and AQ score: those with higher AQ scores, namely stronger autistic-like tendencies and lower social proficiencies, were less biased in making duration judgments of the communicative and the noncommunicative motion sequences, and had PSEs closer to 0 ($r_{90} = -0.40$, p<0.001, *Figure 3B*). The median AQ score of this sample was 19, roughly corresponding to a cut-off between low and intermediate levels of autistic traits (*Baron-Cohen et al., 2001*). A median split of the observers by AQ score showed that the social interaction-induced temporal compression effect was evident in the low AQ group (AQ scores < 20, mean PSE = 58.9 ms, significantly above 0; $t_{45} = 7.06$, p<0.001, Cohen's d = 1.04), yet barely reached statistical significance in the high AQ group (AQ scores ≥ 20, mean PSE = 17.1 ms, $t_{43} = 2.00$, p=0.052), with a significant group difference in PSE ($t_{88} = 3.51$, p=0.001, Cohen's d = 0.74), but not difference limen ($t_{88} = 0.53$, p=0.60) (*Figure 3C*). These results, while reaffirming the influence of perceived social interactions on subjective duration, highlighted the idiosyncrasy of subjective time for social interactions, and tied it to a stable personality trait — social proficiency.

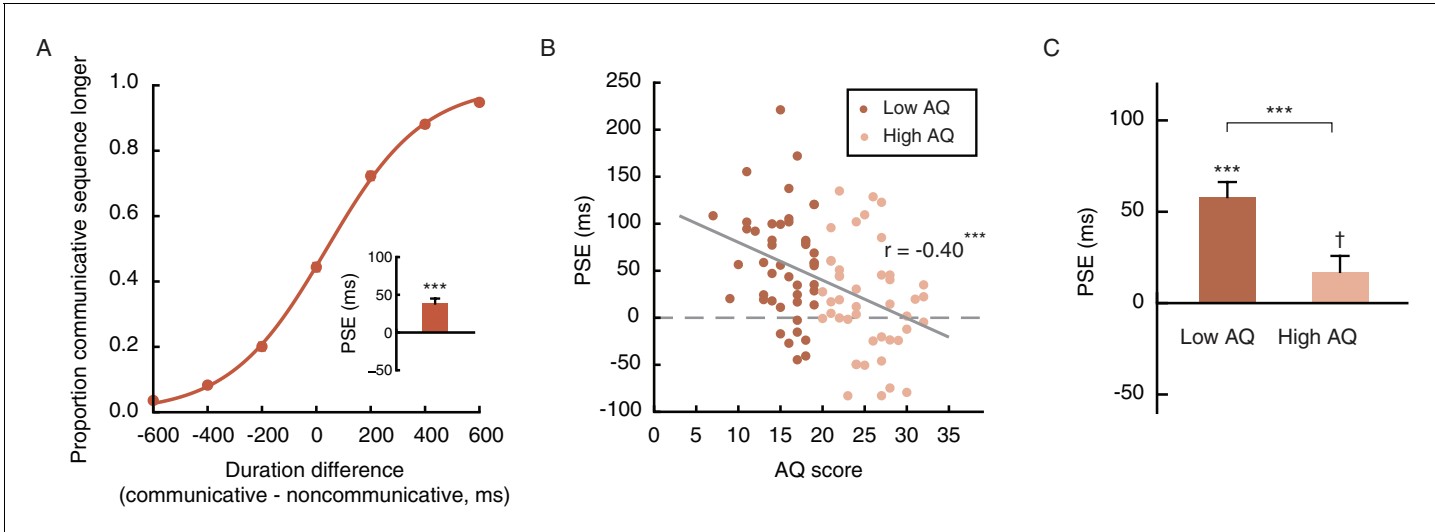

**Figure 3.** The degree of temporal compression induced by the perception of social interactions reflects one's social proficiency. (**A**) Psychometric function for Experiment 4 which contained only the upright condition. Inset shows the overall PSE. (**B**) Across the observers, one's PSE negatively correlated with his score on the Autism Spectrum Quotient (AQ). (**C**) Comparison of the PSEs for low AQ (<20) versus high AQ (≥20) observers. Error bars: standard errors of the mean; †: marginally significant, ***p≤0.001.
DOI: https://doi.org/10.7554/eLife.32100.006

## Oxytocin mediates temporal perception of social interactions

Autistic traits have been widely associated with reduced levels of oxytocin (*Clark et al., 2013*; *Green et al., 2001*; *Modahl et al., 1998*; *Parker et al., 2014*), and can be ameliorated with intranasal oxytocin administration (*Anagnostou et al., 2012*; *Gordon et al., 2013*; *Guastella et al., 2010*; *Watanabe et al., 2015*; *Yatawara et al., 2016*). The link between autistic-like tendency and subjective duration of social interactions thus raises the intriguing question of whether oxytocin plays a role therein. Experiment 5 probed this question by testing if the application of oxytocin would promote the social interaction-induced temporal compression effect in socially less proficient individuals. The same task as in Experiment 4 was employed. Eighty males with AQ scores larger than or equal to 20 (range: 20–36, same cutoff value as in Experiment 4) completed the duration judgment task twice, once before and once 40 min after the nasal administration of either oxytocin (for 40 observers) or atosiban (for the other 40 observers). Atosiban is a desamino-oxytocin analogue and a competitive oxytocin receptor antagonist (*Sanu and Lamont, 2010*), and has been shown to be centrally available when administered intranasally (*Lamont and Kam, 2008*; *Lundin et al., 1986*). We used it as a comparison treatment and hypothesized that its influence on subjective duration of social interactions, if any, would be in the opposite direction of oxytocin. The results were consistent with our hypotheses. In those treated with oxytocin, the mean PSE significantly increased by 36.9 ms ($t_{39} = 3.68$, p=0.001, Cohen's d = 0.58), from 13.3 ms at baseline, which was not significantly different from 0 ($t_{39} = 1.34$, p=0.19), to 50.1 ms after oxytocin administration (significantly above 0, $t_{39} = 4.22$, p<0.001, Cohen's d = 0.66) (*Figure 4A*). By contrast, in those treated with atosiban, the mean PSE was not significantly different from 0 both at baseline (9.7 ms, $t_{39} = 1.14$, p=0.26) and after atosiban administration (−9.4 ms, $t_{39} = −1.11$, p=0.28), yet numerically showed a significant reduction ($t_{39} = −2.24$, p=0.031, Cohen's d = −0.35) (*Figure 4B*). Between the two drug groups, there was a marked difference in the changes in PSEs pre- and post- drug administration ($t_{78} = 4.25$, p<0.001, Cohen's d = 0.95). These effects could not be due to changes of the observers' temporal sensitivity, as their difference limens remained unaltered before and after drug administration ($t_{39}$s = 0.35 and −0.66, ps = 0.73 and 0.51, for oxytocin and atosiban, respectively). Besides, their transient mood states, as reflected by ratings on the Profile of Mood States (POMS) scale (*McNair et al., 1971*), were also unaffected by drug condition (drug condition ×testing session; total mood disturbance: $F_{1, 78} = 0.18$, p=0.68; all subscales: ps > 0.1).

Nonetheless, there is a reason to suspect that the influence of atosiban on socially less proficient individuals, as observed in Experiment 5, was unreliable — they were not significantly biased by the social aspect of interactions in making duration judgments to begin with. To further verify if antagonizing the effect of oxytocin would diminish the temporal compression effect of perceived social interactions, we turned to socially proficient individuals. In Experiment 6, 80 males with AQ scores less than 20 (range: 10–19, same cutoff value as in Experiment 4) completed the duration judgment task both before and 40 min after the nasal administration of either atosiban (for 40 observers) or saline (for the other 40 observers), following the same procedure as in Experiment 5. Saline served as a placebo control here to address potential confounds including practice and fatigue. At baseline, the observers in both drug groups were significantly biased towards perceiving the communicative motion sequences as shorter in duration than the noncommunicative ones (mean PSEs = 48.6 ms and 54.5 ms, $t_{39}$s = 4.32 and 5.94, ps < 0.001, Cohen's ds = 0.68 and 0.94, for atosiban and saline, respectively), with no difference in between ($t_{78} = 0.41$, p=0.68) (*Figure 4C–D*). After drug treatments, however, a significant group difference emerged (mean PSEs = 17.5 ms and 46.8 ms, $t_{78} = 2.64$, p=0.010, Cohen's d = 0.59). In those treated with atosiban, the mean PSE dropped significantly by 31.1 ms ($t_{39} = −3.90$, p<0.001, Cohen's d = −0.62), albeit still significantly above 0 ($t_{39} = 2.04$, p=0.048, Cohen's d = 0.32) (*Figure 4C*). By contrast, in those treated with saline, the PSEs were unaffected ($t_{39} = −1.13$, p=0.26; *Figure 4D*). There was a significant difference between the two drug groups in the changes in PSEs pre- and post- drug administration ($t_{78} = 2.22$, p=0.029, Cohen's d = 0.50). Meanwhile, the difference limens in both groups remained unchanged ($t_{39}$s = −1.07 and −1.11, ps = 0.29 and 0.27, for atosiban and saline, respectively), and the POMS ratings were unaffected by drug condition (drug condition ×testing session; total mood disturbance: $F_{1, 78} = 0.47$, p=0.50; all subscales: ps > 0.2).

We plotted in *Figure 4E* the distributions of PSEs for individual observers in Experiments 5 and 6 before (x-axis) and after (y-axis) drug treatment. Their central tendencies are respectively highlighted

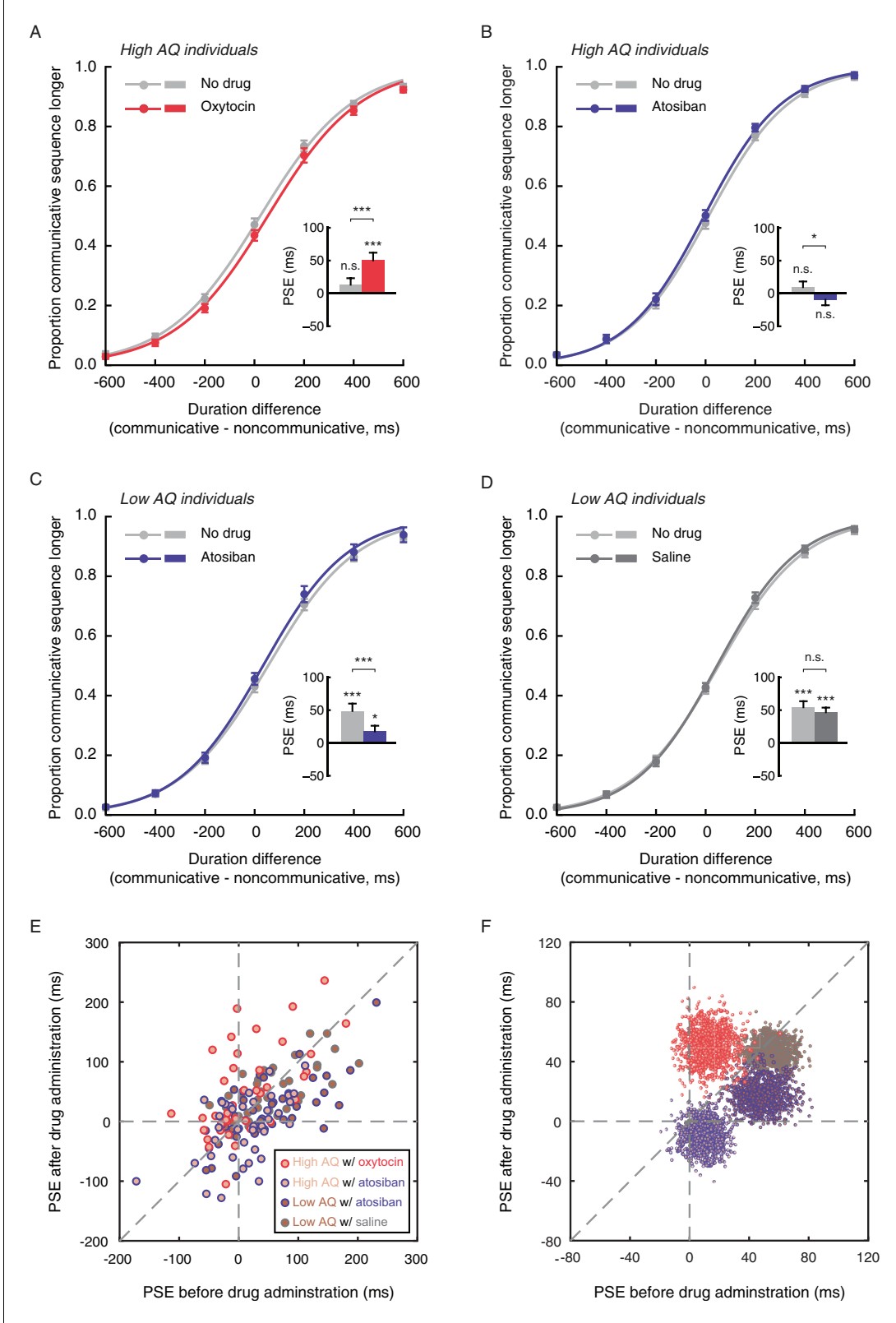

**Figure 4.** Oxytocin mediates temporal perception of social interactions. (A-D) Psychometric functions for Experiments 5 and 6 where high AQ observers (A-B) and low AQ observers (C-D) completed the duration judgment task of upright motion sequences both before (light gray curves) and after the nasal administration of oxytocin (red curve in A), atosiban (blue curves in B and C) or saline (dark gray curve in D). Insets show the PSEs. (E) The PSEs after drug administration versus those before drug administration for high AQ individuals treated with oxytocin (light brown dots with red circles), high

*Figure 4 continued on next page*

*Figure 4 continued*

AQ individuals treated with atosiban (light brown dots with blue circles), low AQ individuals treated with atosiban (dark brown dots with blue circles) and low AQ individuals treated with saline (dark brown dots with gray circles). A slope of 1 (dashed diagonal line) represents comparable PSEs before and after drug administration. (**F**) Bivariate distributions of 1000 bootstrapped sample means for each group. Error bars: standard errors of the mean; *p<0.05, ***p≤0.001.

DOI: https://doi.org/10.7554/eLife.32100.007

in *Figure 4F*, generated by using a standard bootstrapping procedure (*Davison and Hinkley, 1997*). The majority of the observers with a high AQ score that were treated with oxytocin (light brown dots with red circles) fell around the dashed line of x = 0 on the positive side and above the dashed line of slope 1. In other words, they were not significantly biased in making duration judgments of communicative and noncommunicative motion sequences before oxytocin administration but became biased towards perceiving the communicative ones as shorter in duration afterwards, and showed increased PSEs. By contrast, the observers with a low AQ score that were treated with atosiban (dark brown dots with blue circles) mainly fell on the positive side of the dashed line of y = 0 and somewhat above it, but below the dashed line of slope 1. Those with a high AQ score that were treated with atosiban (light brown dots with blue circles) largely fell around the origin and a bit below the dashed line of slope 1. There was not much overlap between the observers treated with oxytocin (dots with red circles) and those treated with atosiban (dots with blue circles). Roughly in between lay the observers treated with saline (low AQ individuals, dark brown dots with gray circles), who mainly fell around the dashed line of slope 1 in the first quadrant of the x-y plane.

## Discussion

As one of many social species, humans seek out companionship and social interactions. The current study demonstrates that such a motive is weaved in our mental representation of time: Motion sequences depicting agents engaging in social interactions tend to be perceived as shorter in duration than those showing agents acting noncommunicatively. This temporal compression effect is independent of the basic visual features or the non-biological properties of the agents (Experiment 1), or the spatial-temporal correlations in between (Experiments 2 and 3), and cannot be explained by the inference of causality or the contingency between two entities' movements (Supplementary Experiment). Rather, it relies on the observer's intrinsic autistic-like tendency, such that socially less proficient individuals are less susceptible to the effect than socially proficient ones (Experiment 4). The mechanism underlying this phenomenon critically involves oxytocin. In socially less proficient individuals that overall have lower levels of endogenous oxytocin (*Andari et al., 2014*; *Modahl et al., 1998*; *Parker et al., 2014*), oxytocin administration promotes the social interaction induced temporal compression effect (Experiment 5). By contrast, in socially proficient individuals with overall higher levels of endogenous oxytocin, the very effect is diminished following the application of an oxytocin antagonist named atosiban (Experiment 6). Whereas distortion of time perception has been widely associated with properties of the stimuli (*Eagleman, 2008*) and the context (*Shi et al., 2013*), these findings provide, to our best knowledge, the first empirical evidence that links the subjective time of an event with a personality trait, namely social proficiency. In doing so, they open up a new avenue to probe automatic processing of complex social interplays at the individual level.

Social perception involves multifaceted information processing that culminates in the accurate recognition of others' dispositions and intensions. It inherently entails the integration of information, including the integration between origin and effect (causal integration), the integration of clues (e.g. motion cues) to form impressions, extract intentions, and arrive at judgments (cognitive algebra), etc (*Anderson, 1981*; *Blythe et al., 1999*; *Heider, 1944*; *Smith, 1984*). Autism spectrum disorders have been associated with deficits in sensory integration (*Brandwein et al., 2013*; *Gepner and Mestre, 2002*), though not with unisensory temporal function per se (*Stevenson et al., 2014*). It is plausible that such deficits cascade into the domain of social interactions and partially cause the aforementioned effects.

Multiple brain regions are engaged in social perception, and coordinately enable efficient assessment and interpretation of social signals. They include not only the STS that supports the

understanding of actions (*Allison et al., 2000*), but also higher cortical areas like the temporal-parietal junction (TPJ) that represents mental states (*Carter and Huettel, 2013*) and the dorsal medial prefrontal cortex (dmPFC) that is implicated in the uniquely human representation of triadic interactions between two minds and an object (*Saxe, 2006*). The observed temporal compression effect, being independent of the perception of biological motion, likely arises from these higher stages of social processing (*Cusack et al., 2015*; *von der Lühe et al., 2016*). Subjective time has been proposed to be a signature of the amount of energy expended in representing a stimulus (*Eagleman and Pariyadath, 2009*). It is plausible that communicative motion sequences are processed with increased efficiency in TPJ and dmPFC relative to motion sequences without a recognizable communicative intention, thus leading to lowered metabolic cost (*Gutnisky and Dragoi, 2008*; *Laughlin, 2001*) and shortened subjective duration, particularly in socially proficient individuals. Of note here is that this temporal compression effect can hardly be accounted for by the operation of a dedicated neural module specialized for representing the temporal relationships between events (*Ivry and Schlerf, 2008*), since it is specific to the perception of social interactions.

Oxytocin partially drives social motivation (*Dölen et al., 2013*) and has increasingly been identified as an important modulator of social behaviors (*Carter, 2014*; *Donaldson and Young, 2008*) ranging from social recognition (*Oettl et al., 2016*) to consolation (*Burkett et al., 2016*) and ethnocentrism (*De Dreu et al., 2011*). It has also been linked to repetitive behaviors — a core feature of autism (*Hollander et al., 2003*; *Insel et al., 1999*). Results from human brain imaging studies indicate that the application of oxytocin modulates responses in the STS, TPJ and the prefrontal cortex, among others, in various social cognitive tasks (*Zink and Meyer-Lindenberg, 2012*). Such alterations of neural activities presumably facilitate social processing in socially less proficient individuals, where exogenous oxytocin has been shown to exert a more definitive prosocial effect (*Bartz et al., 2010*; *Bartz et al., 2011*), thereby giving rise to their shortened subjective duration of social interactions following oxytocin administration. In socially proficient individuals, the influence of exogenous oxytocin is more complicated, even hard to predict and interpret (*Bartz, 2016*; *Bartz et al., 2010*; *Bartz et al., 2011*; *Olff et al., 2013*). Nevertheless, we were able to show that antagonizing the effect of endogenous oxytocin with atosiban in this group of observers reduced the temporal compression effect as compared to a placebo control. The comparison between oxytocin and atosiban spoke directly to the role of oxytocin in mediating time perception of social interactions. This finding also adds to and extends the extant pharmacological research on time perception, which has primarily focused on dopamine, serotonin and acetylcholine (*Allman and Meck, 2012*; *Meck, 1998*).

We conclude that the perceived duration of social interactions is a product of complex neuroendocrine and neural processes, the exact mechanism awaiting further investigation, and is ingrained with one's social traits. Subjective time is not only nonuniform, like Einstein alluded to in his analogy, but also idiosyncratic.

## Materials and methods

### Participants

Seventy-two young adults (39 females; mean age ± SD = 22.8 ± 2.5 years) participated in Experiments 1 to 3, 24 in each experiment. Sample sizes were determined by G*Power to be sufficient to detect a large temporal distortion effect in time perception (d ≥ 0.8), at a power larger than 95%. The effect size was estimated based on an earlier study that employed biological motion stimuli and similar psychophysical testing procedures to those in the current study (*Wang and Jiang, 2012*). An independent group of 90 males (22.5 ± 2.7 years) took part in Experiment 4. In addition, 80 male non-smokers (22.1 ± 2.6 years) who scored above or equal to 20 on the Autism Spectrum Quotient (AQ) (mean AQ score ±SD = 24.9±4.3) participated in Experiment 5. Another 80 male non-smokers (22.8 ± 2.3 years) with AQ scores below 20 (15.5 ± 2.7) participated in Experiment 6. Only males were recruited in Experiments 4–6 for the following reasons: In Experiment 4, this was done to take advantage of men's pronounced variance in social proficiency (*Baron-Cohen et al., 2001*). In Experiments 5 and 6, this was due to pragmatic reasons (like most intranasal oxytocin studies), as oxytocin and atosiban respectively cause and antagonize contractions of the uterus. All participants reported to have normal or corrected-to-normal vision. Those in Experiments 5 and 6 also reported to have no respiratory allergy or upper respiratory infection at the time of testing. They gave informed

consent to participate in procedures approved by the Institutional Review Board at Institute of Psychology, Chinese Academy of Sciences.

## Visual stimuli

Ten communicative point-light motion sequences (C1-10 in *Supplementary file 1*), each portraying two agents of the same gender engaging in either a face-to-face (dyadic, two motion sequences) or a person-object-person (triadic, eight motion sequences) interaction, were chosen from the Communicative Interaction Database (*Manera et al., 2010*). By cross-pairing the agents of the same gender from different interactions, we produced an essentially physically matched set of 10 noncommunicative motion sequences (NC1-10 in *Supplementary file 1*). We verified in an independent group of 24 observers (half male, 27.5 ± 3.3 years) that the noncommunicative motion sequences were perceived as significantly less communicative than the communicative ones (normalized communicativeness rating: 0.37 vs. 0.71, $t_{23} = -8.11$, p<0.0001, Cohen's d = −1.66). These twenty motion sequences and their inverted (upside-down) counterparts were used in Experiment 1, shown at 30 frames per second with a visual angle of approximately 6°×9° (each agent was approximately 2°×9°). In Experiments 2 and 3, a temporally delayed version and a spatially swapped version of the stimuli in Experiment 1 were respectively adopted. Specifically, we introduced a temporal lag of 700 ms (21 frames) in between every two acting agents (communicative and noncommunicative, upright and inverted) in Experiment 2 (*Manera et al., 2013*), and spatially swapped the two agents in each display in Experiment 3 such that they faced in opposite directions instead of facing each other. The original upright communicative and noncommunicative motion sequences were employed in Experiments 4–6.

## Behavioral procedures

Each trial of the duration judgment task in Experiment 1 began with a fixation on a central cross (1°×1°) for 1000 ms, followed by two sequentially presented motion sequences — one communicative, the other noncommunicative, in random order with a blank screen of 400–600 ms in between (*Figure 1*). One of the motion sequences (communicative or noncommunicative, each in half of the trials in random order) was presented for 1000 ms, the other for 400, 600, 800, 1000, 1200, 1400 or 1600 ms with equal possibility. That is, the duration difference between the communicative and the noncommunicative motion sequences ranged from −600 ms to 600 ms in steps of 200 ms. Observers were asked to press one of two buttons to indicate which motion sequence (the first or the second) was longer in duration, a task that did not require recognitions of the nature of the motion sequences. The next trial began immediately after a response was made. There were 35 trials in each block and a total of 8 blocks. In half of the blocks, the motion sequences were presented upright; in the other half, they were presented upside down. The order of the 'upright' and the 'inverted' blocks was balanced across observers. There was a short break of 1 to 2 min in between the blocks.

Experiments 2 to 6 followed the same procedures as in Experiment 1 except for the followings: In Experiments 2 and 3, a temporally delayed version and a spatially swapped version of the visual stimuli were respectively used (see Visual stimuli above). As a result of the temporal lag inserted between the acting agents, each motion sequence in Experiment 2 was presented for 700 ms longer than in Experiment 1 on average. In Experiment 4, observers only completed 4 'upright' blocks. In Experiments 5 and 6, observers completed 4 'upright' blocks at baseline and another 4 'upright' blocks 40 min after drug treatment (see Drug Application below). They also provided ratings on the Profile of Mood States scale (*McNair et al., 1971*) following the duration judgment task both at baseline and after drug treatment.

## Drug application

Observers in Experiments 5 and 6 self-administered a single intranasal dose of 24 IU of oxytocin (ProSpec,>99.0% as determined by RP-HPLC, dissolved in saline; three puffs per nostril, each with 4 IU of oxytocin; for half of the observers in Experiment 5), 60 µg of atosiban (ProSpec,>99.0% as determined by RP-HPLC, dissolved in saline; three puffs per nostril, each with 10 µg of atosiban; for half of the observers in Experiments 5 and 6) or saline (three puffs per nostril, for half of the observers in Experiment 6), in a randomized between-subjects manner, after they completed the baseline

blocks of the duration judgment task and the Profile of Mood States scale. Neither the participants nor the experimenters were aware of the identity of the drug used. Atosiban is a desamino-oxytocin analogue and a competitive oxytocin receptor antagonist (*Sanu and Lamont, 2010*). Both oxytocin (*Dal Monte et al., 2014*; *Freeman et al., 2016*; *Lee and Weerts, 2016*; *Striepens et al., 2013*) and atosiban (*Lamont and Kam, 2008*; *Lundin et al., 1986*) have been shown to be bioavailable when administered intranasally.

Fresh oxytocin and atosiban solutions were made every 3 days during the period of data collection, such that for each participant in the drug administration experiments, the solution he received was prepared in less than 3 days before. The prepared solutions were stored in 10 ml sterilized nasal spray bottles at 4°C until usage.

## Analysis

For each observer under each condition, we calculated the proportions that a communicative motion sequence was judged as longer in duration than a noncommunicative one, and fitted them with a Boltzmann sigmoid function $f(x) = 1/(1 + \exp((x - x_0)/\omega))$, where $x$ corresponds to the physical duration difference between a communicative motion sequence and a noncommunicative one (-600 ms, -400 ms, -200 ms, 0 ms, 200 ms, 400 ms, or 600 ms), $x_0$ corresponds to the point of subjective equality (PSE), at which the observer perceived a communicative motion sequence as equal in duration to a noncommunicative one; and half the interquartile range of the fitted function corresponds to difference limen, an index of temporal discrimination sensitivity. PSE and difference limen served as the dependent variables and were subsequently compared between conditions and groups.

## Acknowledgements

We thank Yuting Ye for assistance, and Zhi Yang and Qian Xu for comments and suggestions. This work was supported by the National Natural Science Foundation of China (31422023 and 31525011), the Strategic Priority Research Program (XDB02010003 and XDB02030006) and the Key Research Program of Frontier Sciences (QYZDB-SSW-SMC030 and QYZDB-SSW-SMC055) of the Chinese Academy of Sciences.

## Additional information

### Author contributions

Rui Liu, Xiangyong Yuan, Kepu Chen, Data curation, Software, Formal analysis, Validation, Investigation; Yi Jiang, Conceptualization, Supervision, Funding acquisition, Methodology, Writing—review and editing; Wen Zhou, Conceptualization, Supervision, Funding acquisition, Methodology, Writing—original draft, Writing—review and editing

### Author ORCIDs

Yi Jiang (iD) https://orcid.org/0000-0002-5746-7301
Wen Zhou (iD) https://orcid.org/0000-0001-6730-2116

### Funding

| Funder | Grant reference number | Author |
| --- | --- | --- |
| National Natural Science Foundation of China | 31525011 | Yi Jiang |
| Chinese Academy of Sciences | XDB02010003 | Yi Jiang |
| Chinese Academy of Sciences | QYZDB-SSW-SMC030 | Yi Jiang |
| National Natural Science Foundation of China | 31422023 | Wen Zhou |
| Chinese Academy of Sciences | XDB02030006 | Wen Zhou |
| Chinese Academy of Sciences | QYZDB-SSW-SMC055 | Wen Zhou |

The funders had no role in study design, data collection and interpretation, or the decision to submit the work for publication.

## Decision letter and Author response

Decision letter https://doi.org/10.7554/eLife.32100.012
Author response https://doi.org/10.7554/eLife.32100.013

## Additional files

### Supplementary files

• Source data 1. Raw data. PSEs and difference limens (DLs) for individual participants in Experiments 1-6.
DOI: https://doi.org/10.7554/eLife.32100.008

• Supplementary file 1. Description of the point-light motion sequences used in the duration judgment task. C1-10 were chosen from the Communicative Interaction Database (*Manera et al., 2010*). NC1-10 were produced by cross-pairing the agents of the same gender from C1-10.
DOI: https://doi.org/10.7554/eLife.32100.009

• Transparent reporting form
DOI: https://doi.org/10.7554/eLife.32100.010

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
