## [Decision Letter]

[Editors’ note: a previous version of this study was rejected after peer review, but the authors submitted for reconsideration. The first decision letter after peer review is shown below.]

Thank you for submitting your work entitled "Perception of social interaction idiosyncratically compresses subjective duration in an oxytocin-dependent manner" for consideration by *eLife*. Your article has been reviewed by three peer reviewers, and the evaluation has been overseen by a Guest Reviewing Editor and a Senior Editor. The following individuals involved in review of your submission have agreed to reveal their identity: Jennifer Bartz (Reviewer #2).

Our decision has been reached after consultation between the reviewers. Based on these discussions and the individual reviews below, we regret to inform you that your work will not be considered further for publication in *eLife*.

The good news is that all the reviewers found the work interesting if not downright fascinating and lauded the use of diverse methods. The results certainly are intriguing and thought provoking. The less good news is that the studies lack the power to be confident that the effects observed are, in fact, robust, and the manuscript would be significantly strengthened by substantially bolstering the sample sizes across the board.

On the one hand, the fact that core effects replicate across different samples of participants, however small, is important. On the other hand, sample sizes of n = 12 per group in nearly all reported studies, especially given the apparently modest effect sizes reported, are not large enough to be confident in the present results. Although one reviewer was willing to overlook this for the drug administration studies, which they recognize as difficult to run with samples of any size, the fact that sample sizes were small across the board – combined with some of the concerns highlighted below – means that in its present form the paper is not appropriate for consideration for publication in *eLife*.

Beyond this core concern, there are a number of issues raised by the reviewers that concern the lack of a single clear interpretation for the findings – and possible alternatives not considered in the paper – as well as the value of control conditions that are not present in the reported studies. For example, one important issue concerns the lack of placebo control for the OXT data. While the DV used here is arguably less susceptible to the kind of expectancy effects that can plague other DVs used in the social cognition and oxytocin literatures, including a placebo group is standard procedure in oxytocin administration studies. The atosiban group is useful here to show that the effects of drug administration differ as a function of the drug, but it would still be useful to have the placebo group as a control to determine the extent to which expectancies impact the observed results. The authors could address this limitation in the discussion.

Also important is the point raised by reviewer #1 concerning the spatial and temporal disruptions in the control stimuli, which impact not only the perception of communicative intentions but may also change the basic perceptual complexity of the task, taxing participants' ability to integrate disparate perceptual signals. As this reviewer notes, a difference in complexity and need to integrate complex cues may crosscut social vs. nonsocial comparisons, and it could be valuable to include a control condition in which a participant interacts contingently with a non-social object (with a parallel disruption of perceptual cues). Given deficits in temporal integration of events over time shown by individuals with ASD (see reference cited by this reviewer), this would help determine whether the present data derive from something social per se as opposed to the ability to form a cohesive and integrated representation of events.

Along these lines, I would highlight the concerns that reviewer #2 raises concerning other possible factors that could mediate the effects of oxytocin. For example, oxytocin has been implicated in obsessive/repetitive/stereotyped behaviors (e.g., Leckman et al., 1994; McDougal et al., 1999; Insel et al., 1999; Hollander et al., 2003) as well as anxiety, which is one of the reasons some have hypothesized that oxytocin may be dysregulated in autism, given that both oxytocin and autism have been linked with repetitive behaviors/rigidity and social impairments. It is possible that the effects of oxytocin observed here could be mediated by these other processes, which should be raised in the discussion.

The bottom line is that bolstering sample sizes and addressing the concerns raised by the reviewers would substantially strengthen the manuscript should you submit it for publication in another venue.

Reviewer #1:

The authors connect an impressively varied subject matter from biological motion, perception of time, autism, and oxytocin to support a very interesting hypothesis about temporal compression in perception. My primary concern is the small sample sizes which constrain confidence in the results. For this reason alone, I cannot recommend publication in *eLife*. I sincerely hope that the authors consider collecting a new, larger set of data, ideally after pre-registering the hypotheses and methods. I think the ideas and experimental designs are sufficiently interesting and innovative to make a strong contribution to the literature assuming well-powered empirical support.

The spatial and temporal disruptions in the control stimuli not only disrupt the perception of communicative intentions but presumably also add a measure of perceptual complexity (e.g., a lack of integration of signals reduces the ability of subjects to engage in higher-order data reduction). This difference in complexity is not just about social vs. nonsocial; indeed, one can imagine doing the same to nonsocial displays in which a person is interacting contingently with another object. This is important because people with autism exhibit deficits in temporal integration of events over time more generally and so it is unclear which piece is critical here -- is it the social piece or is it the ability to form a cohesive integration of events (whether social or nonsocial)? See Stevenson et al., 2014 as an example of this more general deficit in people with ASD.

Reviewer #2:

In this research, the authors studied the phenomenon in which the subjective perception of time is compressed when humans process social information-they argue that this "relativity" in time perception reflects the ease associated with human social cognition. To this end, they show that individual variation in social proficiency (as measured by the Autism Quotient) correlates with subjective time perception, with lower (vs. higher) AQ scorers reporting shorter durations for socially communicative (vs. non-communicative) motion sequences. Moreover, they show that oxytocin, a neurohormone implicated in social affiliation and social cognition, appears to regulate this phenomenon. Specifically, administration of an oxytocin agonist facilitated time compression for less socially proficient individuals, whereas administration of an oxytocin antagonist (atosiban), produced the opposite effect-rendering the experience of time even longer for the socially communicative motion sequences. Finally, in experiment 6, atosiban administration made more socially proficient individuals "look like" less socially proficient individuals in that it lengthened their subjective experience of time when processing social information.

I think this is a fascinating series of studies that has the potential to make a significant contribution to our understanding of human social cognition. I particularly like the diversity of approaches that the authors used to test their hypotheses. This is also the only study I know of that has used both an oxytocin agonist and antagonist to establish the causal role of oxytocin in human social information processing.

Below are questions and comments that arose as I was reviewing this work; since my expertise is in the field of oxytocin, I have focused most of my comments on that aspect of the research.

As noted, I really like the diverse methodological approaches the authors used, and I think that this speaks to the robustness of the phenomenon in question. That having been said, the sample size for the drug administration studies was small, even considering the within-subject design used. Given increased concerns about replication in science, I wondered if the authors could speak to the robustness of the effects. Of course, one option is to collect more data; that said, I appreciate the difficulty and expense of this kind of research. As an alternative, perhaps the authors could use bootstrapping to obtain an estimate of the confidence intervals for the effects observed? In the absence of additional data, it would be reassuring to know that the estimated effect(s) ranged from, say, p=0.01 to p=0.12, and not p=0.01 to p=0.4.

I had a few questions about the oxytocin agonist and antagonist used in this research. First, is "ProSpec" the drug manufacturer? I am not familiar with ProSpec and wondered if the authors could provide some additional information about ProSpec vis a vis quality control. Second, I was not able to access the paper by Lundin et al. showing that atosiban can be administered nasally and, for this reason, thought it might be helpful if the authors could provide more details about Lundin et al's findings regarding the bioavailability of nasally administered atosiban in their paper. Third, it would be helpful if the authors could give more information about the preparation of the atosiban-is this a commercially available product, or was it made by the researchers? What was the concentration of the active ingredient? Do intranasal and IV formulations of atosiban differ and is this relevant for the present investigation?

In Experiments 5 and 6, the researchers preselected participants based on an AQ cutoff of > 20; how was this cutoff determined? I believe the "autism cutoff" on the AQ-adult is 32 (Baron-Cohen et al., 2001). I realize the researchers are not making the claim that their sample is actually on the autism spectrum and, therefore, would not need to use such a rigorous cutoff, but I'm just curious how they selected their cutoff.

The authors might consider spending some time discussing the meaning of the time compression effect in the introduction-that is, why it comes about and what it means. They address this in the discussion (i.e., that it likely reflects "the amount of energy expended in representing a stimulus") but, given the novelty of the phenomena, I think it would be helpful to readers to discuss this a bit in the introduction as well.

Reviewer #3:

This manuscript describes a series of carefully conducted, highly innovative experiments which provide important new information on how social proficiency and oxytocin signaling affect the temporal perception of social interactions in humans. These findings are carefully presented and interpreted and include the following: (1) there is shortened subjective perception of time duration for more socially interactive motion sequences; (2) autistic traits (as measured by the gold-standard AQ) influence subjective social perception, with high AQ individuals failing to show the temporal compression effect; and (3) pharmacological manipulation of the oxytocin system alters temporal compression effects, with oxytocin administration restoring, and an oxytocin antagonist diminishing, this effect in high AQ and low AQ individuals, respectively. The statistical analyses appear appropriate and the methods (with a few minor exceptions) are clearly described. This research study will be of high interest to the broad readership of *eLife*.

1) The lack of a placebo control is problematic. This should be addressed as a study limitation in the Discussion section.

2) The authors in multiple places indicate that endogenous oxytocin levels are associated with measures of social functioning, but they themselves do not test this in their research paradigm. (See also #3 below.) It would be an additional strength of this paper if blood or salivary samples were available and oxytocin levels quantified and analyzed in the context of this report. Are such samples available?

3) Impact statement: The authors did not assess endogenous oxytocin levels, so it is speculative to conclude that these personality traits have neuroendocrine origins.

4) Introduction: I have no idea what this sentence means: "likely through interactions with endogenous oxytocin, intranasally administered oxytocin (Born et al., 2002), is found to alter the processing of.…". Moreover, to my knowledge, there are no studies which have mechanistically evaluated endogenous oxytocin biology in the context of oxytocin administration (in humans or animals) and the Born et al., paper 2002 does not examine oxytocin (but instead vasopressin). I suggest rewriting this sentence for clarity and citing a more accurate reference.

5) Discussion section: "nevertheless, we were able to show that antagonizing endogenous oxytocin" – not technically true; the authors in fact antagonized OXTR.

6) Materials and methods section (subjects): what was the overlap in subjects tested across experiments? Were any repeatedly exposed to the same test stimuli? If so, how did this affect results/was guarded against?

7) Materials and methods section (drug application): What was the latency from drug (agonist and antagonist) administration to behavioral testing? How was the dose of atosiban determined? Were preliminary dose-response studies conducted to determine drug efficacy?

[Editors’ note: what now follows is the decision letter after the authors submitted for further consideration.]

Thank you for submitting your article "Perception of social interaction compresses subjective duration in an oxytocin-dependent manner" for consideration by *eLife*. Your article has been reviewed by three peer reviewers, and the evaluation has been overseen by a Reviewing Editor and Sabine Kastner as the Senior Editor. The following individuals involved in review of your submission have agreed to reveal their identity: Sylvia Morelli (Reviewer #2); Carolyn Parkinson (Reviewer #3).

The reviewers have discussed the reviews with one another and the Reviewing Editor has drafted this decision to help you prepare a revised submission.

As you recall, in the prior submission there were a number of points raised by reviewers with issues of the sample sizes in the studies and the lack of a placebo group in the drug administration studies being among the most significant. We agreed to review this revised manuscript in large part because – in addition to attempting to address the rhetorical and stylistic issues raised in the prior round of reviews – the sample sizes were increased for all studies and the core effects remained significant. With this in mind I was able to secure three reviews for this version of the paper – one from a reviewer of the initial submission (reviewer 1) and two new reviewers. All are experts in the field.

As a group, the reviewers and I all have enthusiasm for the paper's attempt to study an aspect of social perception that is often overlooked – namely, its temporal component. Furthermore, the studies examining the way in which oxytocin impacts such perception, are of great interest.

However, there remain significant questions about the sample size and placebo issues raised in the first round of reviews, and as can be the case when new reviewers bring their fresh perspectives to a manuscript, new and I believe important questions about the nature of the results have been raised. As I believe all comments made by the reviewers are important to address, the individual reviews are appended below.

That said, let me highlight three key issues that all reviewers and myself agree must be addressed in order for this manuscript to receive further consideration:

1) The lack of placebo group, especially in Experiment 6. While you argue that expectancy effects are likely not the cause of the observed results in Experiment 5, It would be useful to add discussion of why issues related to the lack of a placebo control group (beyond expectancy effects), such as practice effects or fatigue, would or would not impact the results in Studies 5 and 6. While we believe that this is less likely to be an issue for Experiment 5, given that half of the high AQ participants were given oxytocin and half were given atosiban, and the results were in opposite directions across groups (i.e., an increase in the PSE in the oxytocin group and a decrease in the atosiban group), it would be more compelling to report whether the changes in PSEs pre- and post-drug administration differed significantly between groups by comparing them directly. An additional question about Study 5 was whether the drugs were administered in a double-blind fashion. I wasn't sure if there was reference to this in the manuscript. Of course, it would be highly problematic if administration was not double blind.

The lack of a placebo-matched control group in Experiment 6 we see as much as more problematic, however, because there is only an atosiban-administration group for the low AQ sample (and no comparable, low AQ group that was administered oxytocin). As such, it is difficult to determine whether the decreased temporal compression effect observed there is due to, for example, practice effects, or to the effects of atosiban administration. Adding in a matched (low AQ) control group that receives a placebo or oxytocin would clarify how to interpret the results of Experiment 6. Indeed, we think this is absolutely critical. A pre-post design with no experimental manipulation does not meet the standards of at a top-tier journal such as *eLife*. We recognize that it's hard to say whether it is feasible to collect such data in two months because addressing this involves more than just running placebo participants – it also means running additional drug participants to ensure random assignment and maintain the double-blind nature of the study.

2) There continue to be questions about sample sizes. Even at their larger sizes (compared to the first submission), the new reviewers were concerned that they were small. All of us felt that power analyses are necessary to justify the current sample sizes. That said, I concur with the first reviewer that the sample sizes for the drug studies are smaller than is the current norm. As this reviewer notes, the effects of oxytocin (and possibly atosiban) can be person-dependent and as such small samples are discouraged to ensure that observed effects are not driven factors that are not of interest, including idiosyncrasies in the way some people respond to oxytocin that may drive effects when sample sizes are small but cancel out when samples are sufficiently large. This reviewer – who is an expert on the administration of intranasal oxytocin – notes that currently studies typically have at least 40-50 participants per cell for between-subject designs and 30 ps for within-subject designs. By contrast, the current Experiments 5 and 6 have much smaller sample sizes per cell. As such, it would be advisable that any new data collected to address the placebo issue include larger samples that honor currently accepted norms.

3) Reviewer 3 raises an important issue concerning the need for a non-social control condition. This reviewer notes that the present Experiments lack non-social control stimuli that use motion displays that control for non-social aspects of motion – e.g. stimuli depicting the interactive, contingent movement of multiple entities/objects (e.g., a person interacting with a tool or object; Michotte-like ball launching stimuli; a stream of dominos falling). There is an important question about whether such displays would be judged to be of shorter duration than comparable displays depicting the same entities moving entirely independently from one another. A revision would need to address this issue as well.

Given the lingering concerns, it would help us to know if you could address the key issues above in a comprehensive and timely manner. Please address your responses in a letter that the reviewers could consider in order to offer additional advice. We are particularly concerned about the time it may take to establish and evaluate a placebo group.

Finally, let me thank you again for submitting this manuscript. We hope you are able to revise it in accord with the above summary – and the more detailed comments of the reviewers below.

Reviewer #1:

I reviewed this paper when it was first submitted to *eLife*. I found the research to be novel and highly interesting. The major criticism with the initial paper was that the sample size for each of the 6 studies was small. The authors have now increased the sample size for all 6 studies, and the effects originally reported hold. Reproducibility is a concern in psychology and other sciences right now and I commend the authors for addressing this in their work-as we're learning, replication, especially when it involves multiple studies, is not always easy (e.g., OSF, Science, 2015).

I was curious how the authors decided what an appropriate sample size would be for each study-was a power analysis conducted? If so, that should be reported. I appreciate the authors' point that establishing the time compression effect does not require a large number of participants because, as they note in response to reviewer 1 – point 1, each participant has a very large number of data points. However, studies 4, 5, and 6 still seem a bit small to me. It is known that the effects of oxytocin (and possibly atosiban) can be person-dependent and, for this reason, small samples are discouraged as the effects could be driven by, e.g., idiosyncrasies in the way some people respond to oxytocin. My impression is that most intranasal oxytocin studies that are published now have at least 40-50 ps/cell for between-subject designs and 30 ps for within-subject designs; with 12 and 24 participants, studies 5 and 6 still seem small to me. Of course, if a power analysis suggests otherwise that would be useful information to know. (Also, the effects are consistent with other work on oxytocin and social cognition so reassuring on that front). The sample size of study 4 also seems a bit small for evaluating the correlation between two variables (e.g., see Lindsay, 2015)-I don't think the authors need a sample as large as 250 (cf. Schönbrodt and Perugini, 2013), since the effect is consistent with prior work linking AQ with indices of social cognitive proficiency, but some justification for the sample size seems necessary given current norms.

In the previous round of reviews concerns were raised about the lack of a placebo control. The authors acknowledged this limitation but argue that expectancy effects were unlikely with the task they used. I agree but note that expectancy effects are not the only reason why a placebo control is useful; other reasons to have a placebo control are to address such confounds as practice effects or fatigue. Is it possible that either of these factors could be driving the effects in studies 5 and/or 6?

Reviewer #2:

Across several experiments, the authors examined the perception of time during social interactions and its relationship to social proficiency. I found the use of drug administration to be novel and a nice addition to the initial experiments. The methods and results were also clearly written and easy to understand. My main concerns surround the theoretical importance of these findings and the small sample sizes.

In both the Introduction and Discussion section, it is unclear why it's important to perceive time accurately in social actions. The paper would be greatly improved if the reader understood more about the perception of time and the social consequences of the temporal compression effect. I didn't have any sense of how this would impact high AQ individuals in everyday life. Although it is clear that there is a temporal compression effect occurring, it is critical to provide more background and theoretical setup for the reader. For example, the authors state, "For instance, intense and/or moving stimuli are generally experienced as longer in duration (Fraisse, 1984) as they evoke stronger perceptual responses in cortical neurons." So, are people with high autism quotient experiencing stimuli as less intense? Or are socially proficient people more engaged by social stimuli and are therefore perceiving them as shorter in duration? What is the proposed mechanism?

Related to the point above, it was unclear why oxytocin should be involved in temporal sequencing. Instead, the Introduction just points out that it should impact the perception of social stimuli & biological motion. This is somewhat vague, so it is also difficult to interpret what we learn from the observed effects of oxytocin.

It would also be helpful to have more background about the development and validation of the stimuli. The noncommunicative motion sequences were created by cross-pairing the agents from different interactions. Did the authors confirm that these sequences were perceived as noncommunicative by the participants? This seems critical to confirm since the conclusions are based on this distinction.

The sample sizes seem small for all studies. The authors should justify why they chose this sample size and demonstrate it has adequate power for detecting the effect. In addition, some samples had female and male participants, while others had only male participants. The authors should justify these decisions and gender as a covariate in their analyses. Lastly, it was unclear why there was no low AQ group for oxytocin administration in Experiment 5.

Reviewer #3:

This article takes a creative and thoughtful approach to testing if the social nature of observed actions, as well as the social proficiency of the perceiver, impact the subjective duration of those actions. Biological motion displays of two individuals were judged to be shorter when an intact communicative interaction was depicted, rather than when footage of individuals from two different interactions was combined into a single display of two individuals acting in parallel but independently from one another; these effects disappeared when the perception of a communicative interaction was interfered with via spatial and temporal manipulations of the stimuli. Additionally, the relative extent of the temporal compression effect for intact communicative displays was stronger among males with lower AQ scores, and the administration of oxytocin strengthened the effect, whereas an oxytocin agonist attenuated the temporal compression effect. The methods used are innovative and the research question is novel and likely to be of interest to a wide audience.

My primary concern is that the "social" (i.e., intact communicative) displays differ from the various spatially and temporally perturbed variations that were tested, as well as from the non-communicative displays, not just because they depict intact social communication, but also because they depict coherent and contingent series of actions that are likely easier for participants to follow, predict and make sense of, compared with the non-communicative displays or the spatially or temporally disrupted displays.

This would seem to be an alternative possible explanation for the temporal compression effects reported here. Would non-social motion displays, that depict the interactive, contingent movement of multiple entities/objects (e.g., a person interacting with a tool or object; Michotte-like ball launching stimuli; a stream of dominos falling) be judged as shorter in duration than comparable displays depicting the same entities moving entirely independently from one another? In other words, is the temporal compression effect observed here reflective of the social nature of the intact communicative displays or of these PL displays being relatively easier for participants to follow, predict and integrate into a single coherent action representation? This seems especially relevant, given that the authors note that the subjective perception of duration is thought to be related to ease of processing. In the absence of a non-social control condition testing for a temporal compression effect for non-social contingent vs. independent motion displays, it would be informative if the authors were to consider this possibility in more detail and discuss the need for future research using analogous non-social stimulus conditions.

---

## [Author Response]

[Editors’ note: what now follows is the decision letter after the authors submitted for further consideration.]

The good news is that all the reviewers found the work interesting if not downright fascinating and lauded the use of diverse methods. The results certainly are intriguing and thought provoking. The less good news is that the studies lack the power to be confident that the effects observed are, in fact, robust, and the manuscript would be significantly strengthened by substantially bolstering the sample sizes across the board.On the one hand, the fact that core effects replicate across different samples of participants, however small, is important. On the other hand, sample sizes of n = 12 per group in nearly all reported studies, especially given the apparently modest effect sizes reported, are not large enough to be confident in the present results. Although one reviewer was willing to overlook this for the drug administration studies, which they recognize as difficult to run with samples of any size, the fact that sample sizes were small across the board – combined with some of the concerns highlighted below – means that in its present form the paper is not appropriate for consideration for publication in eLife.

We appreciate your and the reviewers’ constructive suggestions and overall positive assessment of our work. We have now increased the sample sizes for Experiments 1-6 from ns = 12, 12, 12, 40, 24 and 12, to ns = 24, 24, 24, 50, 48 and 24, respectively, and have successfully replicated our original findings. The new data are now incorporated in the revised manuscript. Please also refer to our detailed response to reviewer 1’s point 1 below.

Beyond this core concern, there are a number of issues raised by the reviewers that concern the lack of a single clear interpretation for the findings – and possible alternatives not considered in the paper – as well as the value of control conditions that are not present in the reported studies. For example, one important issue concerns the lack of placebo control for the OXT data. While the DV used here is arguably less susceptible to the kind of expectancy effects that can plague other DVs used in the social cognition and oxytocin literatures, including a placebo group is standard procedure in oxytocin administration studies. The atosiban group is useful here to show that the effects of drug administration differ as a function of the drug, but it would still be useful to have the placebo group as a control to determine the extent to which expectancies impact the observed results. The authors could address this limitation in the discussion.

This limitation is discussed on Discussion section of the revised manuscript. Please also refer to our detailed response to reviewer 3’s point 1 below.

Also important is the point raised by reviewer #1 concerning the spatial and temporal disruptions in the control stimuli, which impact not only the perception of communicative intentions but may also change the basic perceptual complexity of the task, taxing participants' ability to integrate disparate perceptual signals. As this reviewer notes, a difference in complexity and need to integrate complex cues may crosscut social vs. nonsocial comparisons, and it could be valuable to include a control condition in which a participant interacts contingently with a non-social object (with a parallel disruption of perceptual cues). Given deficits in temporal integration of events over time shown by individuals with ASD (see reference cited by this reviewer), this would help determine whether the present data derive from something social per se as opposed to the ability to form a cohesive and integrated representation of events.

Please refer to our response to reviewer 1’s point 2 below. In brief, based on models of social perception and social inference (Anderson, 1981; Blythe, Todd, and Miller, 1999; Heider, 1944; Smith, 1984), the integration of signals is an integral part of social processing. It is plausible that deficits therein cascade into the domain of social interactions. However, our results were unlikely confounded by differences in the temporal integration of visual events per se. Stevenson et al., showed that people with autism had deficits in multisensory, but not unisensory (visual or auditory), temporal function (Stevenson et al., 2014), whereas our study employed only visual stimuli. Furthermore, we tested healthy young adults without autism, and the high AQ and low AQ participants did not differ in sensitivities of temporal perception. These are now discussed in the Discussion section of the revised manuscript.

Along these lines, I would highlight the concerns that reviewer #2 raises concerning other possible factors that could mediate the effects of oxytocin. For example, oxytocin has been implicated in obsessive/repetitive/stereotyped behaviors (e.g., Leckman et al., 1994; McDougal et al., 1999; Insel et al., 1999; Hollander et al., 2003) as well as anxiety, which is one of the reasons some have hypothesized that oxytocin may be dysregulagted in autism, given that both oxytocin and autism have been linked with repetitive behaviors/rigidity and social impairments. It is possible that the effects of oxytocin observed here could be mediated by these other processes, which should be raised in the Discussion section.

We have briefly discussed this possibility on Discussion section of the revised manuscript.

The bottom line is that bolstering sample sizes and addressing the concerns raised by the reviewers would substantially strengthen the manuscript should you submit it for publication in another venue.Reviewer #1:The authors connect an impressively varied subject matter from biological motion, perception of time, autism, and oxytocin to support a very interesting hypothesis about temporal compression in perception.

We thank the reviewer for his/her positive assessment of our work.

1) My primary concern is the small sample sizes which constrain confidence in the results. For this reason alone, I cannot recommend publication in eLife. I sincerely hope that the authors consider collecting a new, larger set of data, ideally after pre-registering the hypotheses and methods. I think the ideas and experimental designs are sufficiently interesting and innovative to make a strong contribution to the literature assuming well-powered empirical support.

We would like to note that we measured full psychometric functions at seven levels of duration difference between communicative and noncommunicative motion sequences. In the original figures, each dot on a psychophysical curve represented the average of at least 240 data points (20 data points per subject × at least 12 subjects). The mean PSE, which was statistically tested, characterized the entire curve, namely at least 1680 data points. Psychophysical experiments of this type of design are sensitive in quantifying the relationship between physical stimuli and the perception they produce and do not typically require many subjects. For instance, a landmark study in the field of biological motion, which reported adaptation of gender derived from biological motion, had a total of 13 subjects, 7 in Experiment 1 and 6 in Experiment 2 (Jordan et al., 2006).

Nonetheless, we agree with the reviewer that bolstering sample sizes would strengthen the study. We have since increased the sample sizes for Experiments 1-6 from ns = 12, 12, 12, 40, 24 and 12, to ns = 24, 24, 24, 50, 48 and 24, respectively, and have successfully replicated the original findings. The new data are now incorporated in the revised manuscript.

2) The spatial and temporal disruptions in the control stimuli not only disrupt the perception of communicative intentions but presumably also add a measure of perceptual complexity (e.g., a lack of integration of signals reduces the ability of subjects to engage in higher-order data reduction). This difference in complexity is not just about social vs. nonsocial; indeed, one can imagine doing the same to nonsocial displays in which a person is interacting contingently with another object. This is important because people with autism exhibit deficits in temporal integration of events over time more generally and so it is unclear which piece is critical here -- is it the social piece or is it the ability to form a cohesive integration of events (whether social or nonsocial)? See Stevenson et al., 2014 as an example of this more general deficit in people with ASD.

We thank the reviewer for raising this thoughtful point. Social perception and social inference have long been theorized as processes of integration, including the integration between origin and effect (causal integration), the integration of information (e.g. motion cues) to form impressions, extract intentions, arrive at judgments (cognitive algebra), etc. (Anderson, 1981; Blythe et al., 1999; Heider, 1944; Smith, 1984). Even simple social perception like that of biological motion requires the integration between motion and form (Allison, Puce, and McCarthy, 2000), and people with autism show a deficit in biological motion perception (Blake et al., 2003). Hence, the integration of signals seems an integral part of social processing. It is plausible that deficits therein cascade into the domain of social interactions. As an aside, the perception of a person interacting with another object entails causal integration and the extraction of intentions and is arguably social in nature.

With respect to temporal integration per se, Stevenson et al., showed that people with autism had deficits in multisensory, but not unisensory (visual or auditory), temporal function (Stevenson et al., 2014), whereas our study employed only visual stimuli. Furthermore, we tested healthy young adults without autism, and the high AQ and low AQ participants did not differ in sensitivities of temporal perception, as indicated by their difference limens (Experiment 4: t48 = -1.26, p = 0.21; Experiment 5-6: t70 = 0.21, p = 0.84). It is therefore unlikely that our results were confounded by differences in the integration of visual events over time.

These are now incorporated in the Discussion section of the revised manuscript.

Reviewer #2:In this research, the authors studied the phenomenon in which the subjective perception of time is compressed when humans process social information-they argue that this "relativity" in time perception reflects the ease associated with human social cognition. To this end, they show that individual variation in social proficiency (as measured by the Autism Quotient) correlates with subjective time perception, with lower (vs. higher) AQ scorers reporting shorter durations for socially communicative (vs. non-communicative) motion sequences. Moreover, they show that oxytocin, a neurohormone implicated in social affiliation and social cognition, appears to regulate this phenomenon. Specifically, administration of an oxytocin agonist facilitated time compression for less socially proficient individuals, whereas administration of an oxytocin antagonist (atosiban), produced the opposite effect-rendering the experience of time even longer for the socially communicative motion sequences. Finally, in experiment 6, atosiban administration made more socially proficient individuals "look like" less socially proficient individuals in that it lengthened their subjective experience of time when processing social information.I think this is a fascinating series of studies that has the potential to make a significant contribution to our understanding of human social cognition. I particularly like the diversity of approaches that the authors used to test their hypotheses. This is also the only study I know of that has used both an oxytocin agonist and antagonist to establish the causal role of oxytocin in human social information processing.

We appreciate the reviewer’s positive assessment of our work.

Below are questions and comments that arose as I was reviewing this work; since my expertise is in the field of oxytocin, I have focused most of my comments on that aspect of the research.1) As noted, I really like the diverse methodological approaches the authors used, and I think that this speaks to the robustness of the phenomenon in question. That having been said, the sample size for the drug administration studies was small, even considering the within-subject design used. Given increased concerns about replication in science, I wondered if the authors could speak to the robustness of the effects. Of course, one option is to collect more data; that said, I appreciate the difficulty and expense of this kind of research. As an alternative, perhaps the authors could use bootstrapping to obtain an estimate of the confidence intervals for the effects observed? In the absence of additional data, it would be reassuring to know that the estimated effect(s) ranged from, say, p=0.01 to p=0.12, and not p=0.01 to p=0.4.

We appreciate the constructive suggestions. This point is similar to reviewer 1’s point 1. Please refer to our response to that point above. In brief, we have increased the sample sizes for Experiments 1-6 from ns = 12, 12, 12, 40, 24 and 12, to ns = 24, 24, 24, 50, 48 and 24, respectively, and have successfully replicated the original findings. The new data are now incorporated in the revised manuscript.

2) I had a few questions about the oxytocin agonist and antagonist used in this research. First, is "ProSpec" the drug manufacturer? I am not familiar with ProSpec and wondered if the authors could provide some additional information about ProSpec vis a vis quality control.

We ordered oxytocin (>99.0% as determined by RP-HPLC) and atosiban (>99.0% as determined by RP-HPLC) from a local distributor for ProSpec (https://www.prospecbio.com), a biotech company based in Rehovot Israel that provides highly purified proteins.

Second, I was not able to access the paper by Lundin et al. showing that atosiban can be administered nasally and, for this reason, thought it might be helpful if the authors could provide more details about Lundin et al's findings regarding the bioavailability of nasally administered atosiban in their paper.

Lundin et al., found that after intranasal administration of atosiban (100 nmol/kg/body weight), the bioavailability was 10.5 +/- 2.9%. Peak concentrations in plasma appeared 10-45 min after intranasal administration. Moreover, at the end of an observation period of 2 h, measurable amounts in plasma were still found in seven of the twelve intranasal experiments (Lundin et al., 1986). See also (Lamont and Kam, 2008) for a review.

Third, it would be helpful if the authors could give more information about the preparation of the atosiban-is this a commercially available product, or was it made by the researchers?

We prepared the oxytocin and atosiban nasal sprays in the lab. Specifically, we dissolved oxytocin and atosiban, respectively, in saline, and stored the solutions in 10ml sterilized nasal spray bottles at 4 °C. Fresh solutions were made every 3 days during the period of data collection, such that for each participant in the drug administration experiments, the solution he received was made in less than 3 days before.

What was the concentration of the active ingredient?

The concentration of the atosiban solution was 80 μg/ml. 6 puffs of nasal spray contained approximately 0.75 ml of liquid and 60 μg of atosiban. See also our response to a related point (point 7) raised by reviewer 3 below.

Do intranasal and IV formulations of atosiban differ and is this relevant for the present investigation?

As mentioned in our response to this reviewer’s point 2, the atosiban nasal spray used in our study contained only atosiban and saline (60 μg/0.75 ml). The IV formulation of atosiban, Tractocile (https://www.medicines.org.uk/emc/medicine/4297), contains atosiban (as acetate, 6.75 mg/0.9ml), mannitol, hydrochloric acid 1M, and water. This difference is unlikely relevant for the present study, as the oxytocin and atosiban nasal sprays were prepared in the same manner.

3) In Experiments 5 and 6, the researchers preselected participants based on an AQ cutoff of > 20; how was this cutoff determined? I believe the "autism cutoff" on the AQ-adult is 32 (Baron-Cohen et al., 2001). I realize the researchers are not making the claim that their sample is actually on the autism spectrum and, therefore, would not need to use such a rigorous cutoff, but I'm just curious how they selected their cutoff.

As mentioned in the Results section of the original manuscript, the median AQ score of the participants in Experiment 4 (all males) was 20, corresponding to a cut-off between low and intermediate levels of autistic traits (Baron-Cohen et al., 2001). A median split of these participants by AQ score showed that the social interaction induced temporal compression effect was evident in the low AQ group (AQ scores < 20), but not in the high AQ group (AQ scores ≥ 20), with a significant group difference in PSE. The AQ cutoff of 20 was subsequently adopted in Experiments 5 and 6. We have made this explicit in subsection “Oxytocin Mediates Temporal Perception of Social Interactions”of the revised manuscript.

4) The authors might consider spending some time discussing the meaning of the time compression effect in the introduction-that is, why it comes about and what it means. They address this in the discussion (i.e., that it likely reflects "the amount of energy expended in representing a stimulus") but, given the novelty of the phenomena, I think it would be helpful to readers to discuss this a bit in the introduction as well.

Suggestion well taken. We have incorporated a brief description of the neural meaning of subjective time in the Introduction section of the revised manuscript.

Reviewer #3:This manuscript describes a series of carefully conducted, highly innovative experiments which provide important new information on how social proficiency and oxytocin signaling affect the temporal perception of social interactions in humans. These findings are carefully presented and interpreted and include the following: (1) there is shortened subjective perception of time duration for more socially interactive motion sequences; (2) autistic traits (as measured by the gold-standard AQ) influence subjective social perception, with high AQ individuals failing to show the temporal compression effect; and (3) pharmacological manipulation of the oxytocin system alters temporal compression effects, with oxytocin administration restoring, and an oxytocin antagonist diminishing, this effect in high AQ and low AQ individuals, respectively. The statistical analyses appear appropriate and the methods (with a few minor exceptions) are clearly described. This research study will be of high interest to the broad readership of eLife.

We appreciate the reviewer’s positive remarks.

1) The lack of a placebo control is problematic. This should be addressed as a study limitation in the Discussion section.

We did not include a placebo control condition due to the following considerations:

a) Generally, a placebo is used to control for the subject-expectancy effect. The current study employed a two-interval forced choice task where participants judged which of the two intervals appeared longer in duration. The nature of this task made it less susceptible to the influence of a general expectation that drug administration would produce a certain effect.

b) The comparison between an oxytocin agonist (oxytocin) and an oxytocin antagonist (atosiban) speaks directly to the role of oxytocin in mediating subjective temporal perception of social interactions.

Whereas we do not think that the lack of a placebo control compromises the validity of our findings, we agree with the reviewer that a placebo control would be helpful to determine the extent to which expectancies influence the observed results. This limitation is discussed in the Discussion section of the revised manuscript.

2) The authors in multiple places indicate that endogenous oxytocin levels are associated with measures of social functioning, but they themselves do not test this in their research paradigm. (See also #3 below.) It would be an additional strength of this paper if blood or salivary samples were available and oxytocin levels quantified and analyzed in the context of this report. Are such samples available?

Unfortunately, not in the context of this report. We did find that plasma oxytocin level negatively correlates with AQ score in a sample of 48 males unrelated to the current study, consistent with the documented associations between endogenous oxytocin levels and prosocial traits (Andari et al., 2014; Parker et al., 2014). As an aside, it has been argued that oxytocin is not a valid biomarker when measured in saliva by immunoassay (Horvat-Gordon et al., 2005).

3) Impact statement: The authors did not assess endogenous oxytocin levels, so it is speculative to conclude that these personality traits have neuroendocrine origins.

As mentioned above in our response to this reviewer’s point 2, data from our lab and others indicate that social traits have neuroendocrine origins. Since we did not assess endogenous oxytocin levels in the current study, we have changed the expression to “…personality traits, which likely have neuroendocrine origins as per previous research”.

4) Introduction: I have no idea what this sentence means: "likely through interactions with endogenous oxytocin, intranasally administered oxytocin (Born et al., 2002), is found to alter the processing of.…". Moreover, to my knowledge, there are no studies which have mechanistically evaluated endogenous oxytocin biology in the context of oxytocin administration (in humans or animals) and the Born et al., paper 2002 does not examine oxytocin (but instead vasopressin). I suggest rewriting this sentence for clarity and citing a more accurate reference.

As the effect of intranasal oxytocin varied based on subjects’ blood oxytocin concentration (Parker et al., 2017) as well as social proficiency (Bartz, 2016; Bartz et al., 2010; Bartz et al., 2011; Olff et al., 2013), which in turn has been linked with endogenous oxytocin levels (Andari et al., 2014; Parker et al., 2014) (see also our response to this reviewer’s point 2), it can be inferred that exogenous oxytocin and endogenous oxytocin likely interact to modulate behavior. We have made this clear in the Introduction of the revised manuscript. Per your and reviewer 2’s suggestions, we have cited (Dal Monte et al., 2014; Freeman et al., 2016; Lee and Weerts, 2016; Striepens et al., 2013) instead of the Born et al. paper here, to show that intranasally administered oxytocin are centrally available.

5) Discussion section: "nevertheless, we were able to show that antagonizing endogenous oxytocin" – not technically true; the authors in fact antagonized OXTR.

We thank the reviewer for pointing this out. We have since changed it to “…that antagonizing the effect of endogenous oxytocin…” in the Discussion section of the revised manuscript.

6) Materials and methods section (subjects): what was the overlap in subjects tested across experiments? Were any repeatedly exposed to the same test stimuli? If so, how did this affect results/was guarded against?

Each subject participated in only one experiment, so there was no overlap in subjects tested across experiments. The participants in Experiments 5 and 6 performed the duration judgment task twice, once before and once after drug administration. This unlikely affected the reported results, as we were primarily interested in the comparison between drug conditions (oxytocin vs. atosiban).

7) Materials and methods section (drug application): What was the latency from drug (agonist and antagonist) administration to behavioral testing?

As mentioned in subsection “Oxytocin Mediates Temporal Perception of Social Interactions” and subsection “Behavioral Procedures.” of the original manuscript, it was 40 minutes.

How was the dose of atosiban determined?

24 IU (dose of oxytocin used in the current study and several other studies (e.g. (Bartz et al., 2010))) is the equivalent of about 48 μg oxytocin and 60 μg vasopressin. Atosiban is close to oxytocin and vasopressin in both structure and molar mass (994.2, 1007.2, and 1084.2 g/mol, respectively), yet the IU for atosiban has not been established. To be conservative, we used 60 μg atosiban.

Were preliminary dose-response studies conducted to determine drug efficacy?

Atosiban was primarily used as a comparison treatment for oxytocin to qualify the role of oxytocin in mediating subjective temporal perception of social interactions. Since previous studies have shown that intranasally administered atosiban is centrally available (Lamont and Kam, 2008; Lundin et al., 1986) (see also our response to reviewer 2’s point 2), we did not conduct dose-response studies to determine drug efficacy.

[Editors' note: the author responses to the re-review follow.]

As you recall, in the prior submission there were a number of points raised by reviewers with issues of the sample sizes in the studies and the lack of a placebo group in the drug administration studies being among the most significant. We agreed to review this revised manuscript in large part because – in addition to attempting to address the rhetorical and stylistic issues raised in the prior round of reviews – the sample sizes were increased for all studies and the core effects remained significant. With this in mind I was able to secure three reviews for this version of the paper – one from a reviewer of the initial submission (reviewer 1) and two new reviewers. All are experts in the field.As a group, the reviewers and I all have enthusiasm for the paper's attempt to study an aspect of social perception that is often overlooked – namely, its temporal component. Furthermore, the studies examining the way in which oxytocin impacts such perception, are of great interest.However, there remain significant questions about the sample size and placebo issues raised in the first round of reviews, and as can be the case when new reviewers bring their fresh perspectives to a manuscript, new and I believe important questions about the nature of the results have been raised. As I believe all comments made by the reviewers are important to address, the individual reviews are appended below.That said, let me highlight three key issues that all reviewers and myself agree must be addressed in order for this manuscript to receive further consideration:1) The lack of placebo group, especially in Experiment 6. While you argue that expectancy effects are likely not the cause of the observed results in Experiment 5, It would be useful to add discussion of why issues related to the lack of a placebo control group (beyond expectancy effects), such as practice effects or fatigue, would or would not impact the results in Studies 5 and 6. While we believe that this is less likely to be an issue for Experiment 5, given that half of the high AQ participants were given oxytocin and half were given atosiban, and the results were in opposite directions across groups (i.e., an increase in the PSE in the oxytocin group and a decrease in the atosiban group), it would be more compelling to report whether the changes in PSEs pre- and post-drug administration differed significantly between groups by comparing them directly.

Between the two drug groups (oxytocin vs. atosiban) in Experiment 5, there was a marked difference in the changes in PSEs pre- and post- drug administration (t_78_ = 4.25, p < 0.001, Cohen’s d = 0.95). This is now incorporated in subsection “Oxytocin Mediates Temporal Perception of Social Interactions Autistic traits” of the revised main text.

An additional question about Study 5 was whether the drugs were administered in a double-blind fashion. I wasn't sure if there was reference to this in the manuscript. Of course, it would be highly problematic if administration was not double blind.

Yes. Neither the participants nor the experimenters were aware of the identity of the drug used. We have made it explicit in the revised Materials and methods section of the revised main text.

The lack of a placebo-matched control group in Experiment 6 we see as much as more problematic, however, because there is only an atosiban-administration group for the low AQ sample (and no comparable, low AQ group that was administered oxytocin). As such, it is difficult to determine whether the decreased temporal compression effect observed there is due to, for example, practice effects, or to the effects of atosiban administration. Adding in a matched (low AQ) control group that receives a placebo or oxytocin would clarify how to interpret the results of Experiment 6. Indeed, we think this is absolutely critical. A pre-post design with no experimental manipulation does not meet the standards of at a top-tier journal such as eLife. We recognize that it's hard to say whether it is feasible to collect such data in two months because addressing this involves more than just running placebo participants – it also means running additional drug participants to ensure random assignment and maintain the double-blind nature of the study.

Please refer to our responses to reviewer 1’s point 3 and reviewer 2’s point 6 below. In brief, we have included a placebo control (saline) group and tested additional atosiban participants in the revised Experiment 6. We found that, different from atosiban, the administration of saline did not significantly alter the observers’ PSEs (t_39_ = -1.13, p = 0.26). Moreover, between the two drug groups, there was a significant difference in the changes in PSEs pre- and post- drug administration (t_78_ = 2.22, p = 0.029, Cohen’s d = 0.50). We therefore concluded that the decreased temporal compression effect observed in the atosiban group was due to the administration of atosiban instead of potential confounds like practice or fatigue.

2) There continue to be questions about sample sizes. Even at their larger sizes (compared to the first submission), the new reviewers were concerned that they were small. All of us felt that power analyses are necessary to justify the current sample sizes. That said, I concur with the first reviewer that the sample sizes for the drug studies are smaller than is the current norm. As this reviewer notes, the effects of oxytocin (and possibly atosiban) can be person-dependent and as such small samples are discouraged to ensure that observed effects are not driven factors that are not of interest, including idiosyncrasies in the way some people respond to oxytocin that may drive effects when sample sizes are small but cancel out when samples are sufficiently large. This reviewer – who is an expert on the administration of intranasal oxytocin – notes that currently studies typically have at least 40-50 participants per cell for between-subject designs and 30 ps for within-subject designs. By contrast, the current Experiments 5 and 6 have much smaller sample sizes per cell. As such, it would be advisable that any new data collected to address the placebo issue include larger samples that honor currently accepted norms.

In Experiments 1-3 (as well as the new supplementary experiment), the sample sizes of n = 24 per group were determined by G*Power to be sufficient to detect a large temporal distortion effect in time perception (d ≥ 0.8), at a power larger than 95%. The effect size was estimated based on an earlier study that employed biological motion stimuli and similar psychophysical testing procedures to those in the current study (Wang and Jiang, 2012). We note that studies on time perception have commonly obtained large effect sizes (d ≥ 0.8) (Droit-Volet et al., 2010; Frassinetti, Magnani and Oliveri, 2009; Hagura et al., 2012; Matthews, 2015; Sadeghi et al., 2011; Zhou et al., 2014). For Experiments 4-6, we have increased the sample sizes from ns = 50, 48 and 24 to ns = 90, 80 and 80 (40 participants per group), and have replicated and strengthened the original findings. Please refer to our responses to reviewer 1’s points 1 and 2 and reviewer 2’s point 4 below for details.

3) Reviewer 3 raises an important issue concerning the need for a non-social control condition. This reviewer notes that the present Experiments lack non-social control stimuli that use motion displays that control for non-social aspects of motion – e.g. stimuli depicting the interactive, contingent movement of multiple entities/objects (e.g., a person interacting with a tool or object; Michotte-like ball launching stimuli; a stream of dominos falling). There is an important question about whether such displays would be judged to be of shorter duration than comparable displays depicting the same entities moving entirely independently from one another. A revision would need to address this issue as well.

Please refer to our response to reviewer 3’s point 1 below. In brief, we have carried out a supplementary experiment using Michotte-like launching and streaming events (Michotte, 1963). In sharp contrast to the temporal compression effect observed in Experiment 1, we obtained a strong temporal dilation effect (t_23_ = -5.78, p < 0.001, Cohen’s d = -1.2). A “causal” launching event was judged to be of significantly longer duration than a “noncausal” streaming event such that it needed to be 105.8 ms shorter to be perceived as equal in duration to the latter. We thus concluded that the inference of causality, or the contingency between the movements of two entities, is unlikely to account for the temporal compression effect associated with the perception of social interactions.

Reviewer #1:I reviewed this paper when it was first submitted to eLife. I found the research to be novel and highly interesting. The major criticism with the initial paper was that the sample size for each of the 6 studies was small. The authors have now increased the sample size for all 6 studies, and the effects originally reported hold. Reproducibility is a concern in psychology and other sciences right now and I commend the authors for addressing this in their work-as we're learning, replication, especially when it involves multiple studies, is not always easy (e.g., OSF, Science, 2015).

We thank the reviewer for the commendation.

1) I was curious how the authors decided what an appropriate sample size would be for each study-was a power analysis conducted? If so, that should be reported.

The sample sizes of n = 24 per group were determined by G*Power to be sufficient to detect a large temporal distortion effect in time perception (d ≥ 0.8), at a power larger than 95%. The effect size was estimated based on an earlier study that employed biological motion stimuli and similar psychophysical testing procedures to those in the current study (Wang and Jiang, 2012). This is now incorporated in the Materials and methods section of the revised main text. We also note that studies on time perception have commonly obtained large effect sizes (d ≥ 0.8) (Droit-Volet et al., 2010; Frassinetti et al., 2009; Hagura et al., 2012; Matthews, 2015; Sadeghi et al., 2011; Zhou et al., 2014).

2) I appreciate the authors' point that establishing the time compression effect does not require a large number of participants because, as they note in response to reviewer 1 – point 1, each participant has a very large number of data points. However, studies 4, 5, and 6 still seem a bit small to me. It is known that the effects of oxytocin (and possibly atosiban) can be person-dependent and, for this reason, small samples are discouraged as the effects could be driven by, e.g., idiosyncrasies in the way some people respond to oxytocin. My impression is that most intranasal oxytocin studies that are published now have at least 40-50 ps/cell for between-subject designs and 30 ps for within-subject designs; with 12 and 24 participants, studies 5 and 6 still seem small to me. Of course, if a power analysis suggests otherwise that would be useful information to know. (Also, the effects are consistent with other work on oxytocin and social cognition so reassuring on that front).

As mentioned above in our response to this reviewer’s point 1, our power analysis was based on previous studies on time perception. We feel that grouping the participants by their AQ scores likely limits the within-group variance in the effects of oxytocin and atosiban (Bartz, 2016; Bartz et al., 2010; Bartz et al., 2011; Olff et al., 2013). That said, we are unaware of any intranasal oxytocin/atosiban study that has examined time perception with a similar paradigm to that used in the current study. We have thus followed the reviewer’s suggestion and increased the sample sizes of Experiments 5 and 6 from ns = 48 and 24 to ns = 80 and 80 (i.e. from ns = 24 per group to ns = 40 per group, see also our response to this reviewer’s point 3 below), respectively, and have successfully replicated the original findings. The new data are now incorporated in the revised manuscript.

The sample size of study 4 also seems a bit small for evaluating the correlation between two variables (e.g., see Lindsay, 2015)-I don't think the authors need a sample as large as 250 (cf. Schönbrodt and Perugini, 2013), since the effect is consistent with prior work linking AQ with indices of social cognitive proficiency, but some justification for the sample size seems necessary given current norms.

Experiment 4 set out to qualify if the extent of temporal compression induced by the perception of social interactions was related to one’s social proficiency. The goal was not to achieve a stable estimate for the true correlation value ρ, which is the issue dealt with in Schönbrodt and Perugini’s paper (Schönbrodt and Perugini, 2013). Nonetheless, we have increased the sample size from n = 50 to n = 90 and obtained a highly significant correlation, r_90_ = -0.40, p < 0.001. The new data are now incorporated into the Results section of the revised main text.

3) In the previous round of reviews concerns were raised about the lack of a placebo control. The authors acknowledged this limitation but argue that expectancy effects were unlikely with the task they used. I agree but note that expectancy effects are not the only reason why a placebo control is useful; other reasons to have a placebo control are to address such confounds as practice effects or fatigue. Is it possible that either of these factors could be driving the effects in studies 5 and/or 6?

We have now included a placebo control group in Experiment 6, where the observers (n = 40, with AQ scores < 20) similarly completed the duration judgment task twice, once before and once 40 minutes after they self-administered 3 puffs of saline per nostril. At baseline, the observers were significantly biased towards perceiving the communicative motion sequences as shorter in duration than the noncommunicative ones (t_39_ = 5.94, p < 0.001, Cohen’s d = 0.94). They remained so after saline administration (t_39_ = 6.66, p < 0.001, Cohen’s d = 1.05), and showed no significant change in PSE (54.5 ms vs. 46.8 ms, t_39_ = -1.13, p = 0.26) (Figure 4D). By contrast, in those treated with atosiban, the mean PSE dropped significantly by 31.1 ms (t_39_ = -3.90, p < 0.001, Cohen’s d = -0.62). Between the two drug groups, there was a significant difference in the changes in PSEs pre- and post- drug administration (t_78_ = 2.22, p = 0.029, Cohen’s d = 0.50). We thus concluded that the observed effects of oxytocin and atosiban in Experiments 5 and 6 were unlikely driven by confounds like practice or fatigue. The results of the revised Experiment 6 are described in detail in subsection “Oxytocin Mediates Temporal Perception of Social Interactions” of the revised main text.

Reviewer #2:Across several experiments, the authors examined the perception of time during social interactions and its relationship to social proficiency. I found the use of drug administration to be novel and a nice addition to the initial experiments. The methods and results were also clearly written and easy to understand. My main concerns surround the theoretical importance of these findings and the small sample sizes.In both the Introduction and Discussion section, it is unclear why it's important to perceive time accurately in social actions. The paper would be greatly improved if the reader understood more about the perception of time and the social consequences of the temporal compression effect. I didn't have any sense of how this would impact high AQ individuals in everyday life. Although it is clear that there is a temporal compression effect occurring, it is critical to provide more background and theoretical setup for the reader. For example, the authors state, "For instance, intense and/or moving stimuli are generally experienced as longer in duration (Fraisse, 1984) as they evoke stronger perceptual responses in cortical neurons." So, are people with high autism quotient experiencing stimuli as less intense? Or are socially proficient people more engaged by social stimuli and are therefore perceiving them as shorter in duration? What is the proposed mechanism?

We tested healthy young adults and found that the perception of social interactions induced a robust temporal compression effect, particularly in low AQ individuals, and that the extent of the effect reflected one’s social proficiency, a stable personality trait (Experiments 4-6). It is plausible that the lack of this effect partially contributes to the social behaviors manifested by socially less proficient individuals. For instance, they may not enjoy social chitchat as much. That said, we do not feel it causes adverse consequences for these individuals in everyday life on top of their relative lack of social proficiency. Exactly how time perception of social interactions affects one’s social behavior goes beyond the scope of the current study.

As discussed in the Materials and methods section of the original main text, multiple brain regions are engaged in social perception, and coordinately enable efficient assessment and interpretation of social signals. They include not only the superior temporal sulcus (STS) that supports the understanding of actions (Allison, Puce and McCarthy, 2000), but also higher cortical areas like the temporal-parietal junction (TPJ) that represents mental states (Carter and Huettel, 2013) and the dorsal medial prefrontal cortex (dmPFC) that is implicated in the uniquely human representation of triadic interactions between two minds and an object (Saxe, 2006). The observed temporal compression effect, being independent of the perception of biological motion, likely arises from these higher stages of social processing (Cusack, Williams and Neri, 2015; von der Luhe et al., 2016). Subjective time has been proposed to be a signature of the amount of energy expended in representing a stimulus (Eagleman and Pariyadath, 2009). It is plausible that communicative motion sequences are processed with increased efficiency in TPJ and dmPFC relative to motion sequences without a recognizable communicative intention, thus leading to lowered metabolic cost (Gutnisky and Dragoi, 2008; Laughlin, 2001) and shortened subjective duration, particularly in socially proficient individuals.

We have now explicitly stated at the end of the revised Introduction that our hypothesis was that social proficiency would influence the neural efficacy in processing social interactions, and thereby modulate the subjective time of perceived social interactions.

Related to the point above, it was unclear why oxytocin should be involved in temporal sequencing. Instead, the Introduction just points out that it should impact the perception of social stimuli & biological motion. This is somewhat vague, so it is also difficult to interpret what we learn from the observed effects of oxytocin.

We hypothesized that exogenous manipulations of oxytocin level would alter the neural efficacy in processing social interactions, and thereby modulate the subjective time of perceived social interactions. We have now made it explicit in the revised Introduction of the revised main text.

As discussed in detail the Discussion section of the original main text, oxytocin has increasingly been identified as an important modulator of social behaviors (Carter, 2014; Donaldson and Young, 2008). Results from human brain imaging studies indicate that the application of oxytocin modulates responses in the STS, TPJ and the prefrontal cortex, among others, in various social cognitive tasks (Zink and Meyer-Lindenberg, 2012). Such alterations of neural activities presumably facilitate social processing in socially less proficient individuals, where exogenous oxytocin has been shown to exert a more definitive prosocial effect (Bartz et al., 2010; Bartz et al., 2011), thereby giving rise to their shortened subjective duration of social interactions following oxytocin administration.

It would also be helpful to have more background about the development and validation of the stimuli. The noncommunicative motion sequences were created by cross-pairing the agents from different interactions. Did the authors confirm that these sequences were perceived as noncommunicative by the participants? This seems critical to confirm since the conclusions are based on this distinction.

An independent group of 24 observers (half male) rated the communicativeness of each of the 20 motion sequences (the original communicative ones and the noncommunicative ones, shown upright) on a visual analogue scale. The noncommunicative motion sequences were perceived to be significantly less communicative than the communicative ones (normalized communicativeness rating: 0.37 vs. 0.71, t_23_ = -8.11, p < 0.0001, Cohen’s d = -1.66). This is now incorporated in the Materials and methods section of the revised main text.

The sample sizes seem small for all studies. The authors should justify why they chose this sample size and demonstrate it has adequate power for detecting the effect.

This point is similar to Reviewer 1’s Points 1 and 2. Please refer to our responses to those points above. In brief, the sample sizes of n = 24 per group (Experiments 1-3 as well as the new supplementary experiment) were determined by G*Power to be sufficient to detect a large temporal distortion effect in time perception (d ≥ 0.8), at a power larger than 95%. The effect size was estimated based on an earlier study that employed biological motion stimuli and similar psychophysical testing procedures to those in the current study (Wang & Jiang, 2012). We have since increased the sample sizes of Experiments 4-6 from ns = 50, 48 and 24 to ns = 90, 80 and 80, and have replicated and strengthened the original findings. The new data are now incorporated in the revised manuscript.

In addition, some samples had female and male participants, while others had only male participants. The authors should justify these decisions and gender as a covariate in their analyses.

Both male and female participants were recruited in Experiments 1-3 (as well as the supplementary experiment) to establish the temporal compression effect associated with the perception of social interactions. We did not use gender as a covariate in our analyses.

Only males were recruited in Experiments 4-6. This was done in Experiment 4 to take advantage of the pronounced variance of social proficiency in males (Baron-Cohen et al., 2001), so as to probe whether social proficiency correlates with the extent of temporal compression induced by the perception of social interactions. Experiments 5 and 6 tested only males for pragmatic reasons (like most intranasal oxytocin studies), as oxytocin and atosiban respectively cause and antagonize contractions of the uterus. These are now clarified in the Materials and methods section of the revised main text.

Lastly, it was unclear why there was no low AQ group for oxytocin administration in Experiment 5.

It has been shown that the effects of exogenous oxytocin are more complicated in socially proficient individuals, even hard to predict and interpret (Bartz, 2016; Bartz et al., 2010; Bartz et al., 2011; Olff et al., 2013)(p. 21 of the original main text). We therefore did not include a low AQ group for oxytocin administration. Instead, we have followed the editor’s and reviewer 1’s suggestions and included a placebo control (saline) group in the revised Experiment 6 (low AQ individuals). Please refer to our responses to the editor’s point 1 and reviewer 1’s point 3 for details.

Reviewer #3:This article takes a creative and thoughtful approach to testing if the social nature of observed actions, as well as the social proficiency of the perceiver, impact the subjective duration of those actions. Biological motion displays of two individuals were judged to be shorter when an intact communicative interaction was depicted, rather than when footage of individuals from two different interactions was combined into a single display of two individuals acting in parallel but independently from one another; these effects disappeared when the perception of a communicative interaction was interfered with via spatial and temporal manipulations of the stimuli. Additionally, the relative extent of the temporal compression effect for intact communicative displays was stronger among males with lower AQ scores, and the administration of oxytocin strengthened the effect, whereas an oxytocin agonist attenuated the temporal compression effect. The methods used are innovative and the research question is novel and likely to be of interest to a wide audience.

We appreciate the reviewer’s positive remarks.

My primary concern is that the "social" (i.e., intact communicative) displays differ from the various spatially and temporally perturbed variations that were tested, as well as from the non-communicative displays, not just because they depict intact social communication, but also because they depict coherent and contingent series of actions that are likely easier for participants to follow, predict and make sense of, compared with the non-communicative displays or the spatially or temporally disrupted displays.This would seem to be an alternative possible explanation for the temporal compression effects reported here. Would non-social motion displays, that depict the interactive, contingent movement of multiple entities/objects (e.g., a person interacting with a tool or object; Michotte-like ball launching stimuli; a stream of dominos falling) be judged as shorter in duration than comparable displays depicting the same entities moving entirely independently from one another? In other words, is the temporal compression effect observed here reflective of the social nature of the intact communicative displays or of these PL displays being relatively easier for participants to follow, predict and integrate into a single coherent action representation? This seems especially relevant, given that the authors note that the subjective perception of duration is thought to be related to ease of processing. In the absence of a non-social control condition testing for a temporal compression effect for non-social contingent vs. independent motion displays, it would be informative if the authors were to consider this possibility in more detail and discuss the need for future research using analogous non-social stimulus conditions.

We thank the reviewer for raising this thoughtful point. We have followed the reviewer’s suggestion and carried out a supplementary experiment using Michotte-like launching and streaming events (Michotte, 1963). In sharp contrast to the temporal compression effect observed in Experiment 1, we obtained a strong temporal dilation effect (t_23_ = -5.78, p < 0.001, Cohen’s d = -1.2). A “causal” launching event needed to be 105.8 ms shorter to be perceived as equal in duration to a “noncausal” streaming event. We therefore concluded that the inference of causality, or the contingency between the movements of two entities, is unlikely to account for the temporal compression effect associated with the perception of social interactions.

The supplementary experiment is mentioned in the Results section and the Discussion section of the revised main text and is illustrated and described in Figure 2—figure supplement 1 and its legend.